# On the Effect of Vibrotactile Stimulation in Essential Tremor

**DOI:** 10.3390/healthcare12040448

**Published:** 2024-02-09

**Authors:** Ariana Moura Cabral, Julio Salvador Lora-Millán, Adriano Alves Pereira, Eduardo Rocon, Adriano de Oliveira Andrade

**Affiliations:** 1Centre for Innovation and Technology Assessment in Health, Postgraduate Program in Electrical and Biomedical Engineering, Faculty of Electrical Engineering, Federal University of Uberlândia, Uberlândia 38400-902, Brazil; adriano.pereira@ufu.br (A.A.P.); adriano@ufu.br (A.d.O.A.); 2Electronic Technology Department, Rey Juan Carlos University, 28922 Madrid, Spain; julio.lora@urjc.es; 3BioRobotics Group, Centre for Automation and Robotics (CAR), CSIC-UPM, 28500 Madrid, Spain; e.rocon@csic.es

**Keywords:** essential tremor, vibrotactile stimulation, gyroscope, wavelet, approximate entropy, frequency analysis

## Abstract

(1) Background: Vibrotactile stimulation has been studied for tremor, but there is little evidence for Essential Tremor (ET). (2) Methods: This research employed a dataset from a previous study, with data collected from 18 individuals subjected to four vibratory stimuli. To characterise tremor changes before, during, and after stimuli, time and frequency domain features were estimated from the signals. Correlation and regression analyses verified the relationship between features and clinical tremor scores. (3) Results: Individuals responded differently to vibrotactile stimulation. The 250 Hz stimulus was the only one that reduced tremor amplitude after stimulation. Compared to the baseline, the 250 Hz and random frequency stimulation reduced tremor peak power. The clinical scores and amplitude-based features were highly correlated, yielding accurate regression models (mean squared error of 0.09). (4) Conclusions: The stimulation frequency of 250 Hz has the greatest potential to reduce tremors in ET. The accurate regression model and high correlation between estimated features and clinical scales suggest that prediction models can automatically evaluate and control stimulus-induced tremor. A limitation of this research is the relatively reduced sample size.

## 1. Introduction

Essential tremor (ET) is a progressive neurological disease that affects millions of individuals worldwide, with a prevalence of 0.9% and a higher occurrence among the elderly (4.5%) [1]. Although its clinical spectrum can also include non-motor signs [2], motor manifestations significantly impair the quality of life and well-being of patients, leading to compromised motor function, increased dependence on daily activities or self-care, heightened social isolation, and an increased vulnerability to depression [3,4,5].

Individuals with ET typically experience tremors in the hands. During the early stages of the condition, tremors manifest unilaterally. This motor sign initially affects one hand, leading to challenges in performing daily activities or fine motor tasks, including holding a glass, using cutlery, and signing one’s name. As the disease advances, tremors in the upper limb worsen bilaterally [3,6]. In addition, other body regions such as the vocal cords, head, neck, and lower limbs might also be affected [7].

In contrast to other movement disorders that exhibit an overlap of motor signs, tremor is the sole motor sign of ET [3]. In this condition, tremor commonly emerges when a limb is maintained in a fixed posture, such as when the arms are outstretched against gravity. Consequently, the hallmark tremor of ET is known as postural tremor. But tremors during voluntary movement are also common. Their frequency varies from 4 to 12 Hz, depending on the body segment that is affected [4,8].

The disabling character of the tremor emphasises the importance of seeking out more personalised treatments and understanding the influence of external inputs on the motor behaviour of individuals with ET. In this direction, studies [5,9,10,11,12,13,14,15] have been carried out to investigate the potential of the biomechanical loads as well as peripheral electrical and mechanical stimuli in the tremor management of a variety of tremor disorders, including ET.

The reasoning behind this is that, despite the limited understanding of the neuropathology of ET and other tremor disorders, evidence suggests a combination of central and peripheral mechanisms underlying the generation and propagation of tremor [10,16]. In particular, some studies propose the involvement of the cerebellar–thalamic–cortical loop, the cortico–ponto–cerebello–thalamic–cortical loop, and the Guillain–Mollaret triangle loop in the neuropathology of ET [17,18], as illustrated in Figure 1.

As these pathways are involved in sensory processing, the integration of sensory–motor information, and the coordination of movement execution, the purpose of employing external inputs would be to introduce disturbances to the peripheral system in an attempt to alter the response of any of the peripheral or central mechanics of tremor for modulating tremor behaviour.

Among studies on external inputs, research focusing on biomechanical loads was pioneering, comprising a substantial volume of investigations [6,12]. In this direction, several studies have described a significant attenuation of tremor when an inertial load is applied to the affected limb, reporting a decrease in the energy and frequency of the mechanical component of the tremor [10,16,19,20]. However, a voluminous apparatus attached to a limb can sometimes create embarrassment or discomfort in social circumstances.

An alternative approach that has been investigated is peripheral electrical stimulation. Despite recent advances in the investigation of electrical stimulation for tremor control, there remains no consensus in the literature regarding how it affects the circuits that trigger tremor and the latency of its effect on tremor. Tremor reduction is thought to be caused by the generation of muscle forces antagonistic to tremor (e.g., when electrical stimulation is applied to muscles) or even by decreasing the excitability of motor neurons that innervate muscles where tremor is manifested (e.g., when electrical stimulation is applied to peripheral nerves) [21].

Electrical stimulation has been found to be an effective treatment for reducing tremors. The United States Food and Drug Administration has just issued approval for the first wearable transcutaneous electrical nerve stimulator, called Cala Trio [5]. One of the most comprehensive studies using this device for individuals with ET [14] found that it led to a decrease in scores on clinical rating scales for tremor in the dominant hand. Additionally, there was an average reduction in the peak power of the tremor. Nevertheless, not all participants exhibited motor improvement. Clinicians observed a reduction in tremors in 68% of patients, and 60% of patients themselves reported experiencing improvement.

On the other hand, the open-label, single-arm design of the study constrains the generalisability of the observed effects of the device, as the administered stimulation is not withheld from trial participants. Although the individuals assessed in the study were followed for an extended three-month period, the study restricts the quantitative assessment of the long-term effects of electrical stimulation, possible stimulus response latency, or potential motor fluctuations over time. This is because the involuntary activity of the participants’ limbs, captured by accelerometers, lasts only 12 s. Furthermore, the study did not assess the motor behaviour of the subjects’ limbs during stimulation.

Electrical stimulation has limitations as well. Despite significant advancements in technological and design aspects of electrical stimulation technologies, hardware malfunctions or the experience of discomfort and pain resulting from stimulation are reasons that can lead users to discontinue the usage of electrical stimulation in real-world scenarios [9,22]. The research conducted by Isaacson and colleagues [14] revealed that 18% of the participants experienced adverse events. These events included skin irritations such as redness, itchiness, and swelling, as well as sores, lesions, electrical burns, significant discomfort, persistent pain caused by stimulation, and the sensation of an electric shock while using the device.

In light of this context, peripheral vibrotactile stimulation may be another interesting alternative to be investigated and to provide the basis for new treatments. This is because it is a type of peripheral stimulation in which mechanical stimuli are applied to the skin and can be detected by vibration-sensitive receptors, such as the mechanoreceptors present in the skin. When stimulated, mechanoreceptors transmit the sensory information to synapse with second-order neurons in the ipsilateral nucleus cuneatus, as shown in Figure 2, which has important projections to the thalamus and the inferior olive [23].

Thus, the hypothesis is that the application of vibratory stimulation (rhythmic and repeated tactile stimulation with a non-adaptable pattern) in the upper limb can produce an intense normal physiological stimulation (inhibitory) in the nucleus cuneatus that may diminish pathological oscillation (e.g., ION oscillation) and, consequently, may reduce tremor observed in patients with ET. This hypothesis is based on the fact that the nucleus cuneatus has significant projections to the thalamus and the inferior olive (Figure 2), coupled with the idea that the abnormal synchronised rhythmic activity of the inferior olive leads to tremor in harmaline models of ET [24].

Nevertheless, there is a lack of studies conducted on the effectiveness of vibrotactile stimulation on ET. Some investigations have demonstrated that peripheral mechanical stimulation can reduce tremor. On the other hand, some studies have demonstrated an increase in tremor following stimulation [15]. This indicates that there is no consensus in the literature regarding the effectiveness of this type of stimulus for movement disorders, specifically ET. Moreover, in these investigations, the mechanisms underlying vibrotactile stimulation in tremor are unclear.

Therefore, this research was motivated by the lack of studies evaluating the effect of vibrotactile stimulation on postural tremor. In particular, this study expands on the analysis initially proposed by Lora-Millán and colleagues [13] in a study designed to investigate the effect of various vibrotactile stimulation patterns on tremor, analysing the dataset provided by the authors. Although the authors reported that mechanical vibration does not consistently decrease tremor in individuals with ET, the involuntary activity in the hand and forearm is not assessed individually but rather as the resultant difference between them.

However, this approach limits the exploration of the effects of vibration on each individual body segment. Thus, in contrast to previous analyses, this study evaluates the individual effects of vibrotactile stimulation on involuntary hand and forearm activity, also considering the axes individually. In order to better understand the effect of vibrotactile stimulation on the dynamics of postural tremor, this study not only assesses its effects on tremor amplitude and energy, but also introduces an analysis of regularity and explores spectral analysis in more depth. Furthermore, it enhances the technique for eliminating trends and stimulation artefacts by employing wavelets.

## 2. Materials and Methods

### 2.1. Ethical Considerations

The database used in this research originated from a previously conducted study that received approval from the local ethics committee at Hospital 12 de Octubre in Madrid, Spain [13].

### 2.2. Participants

The study assessed the tremor in the most affected upper limb (hand and forearm) of 18 individuals with ET. Among the individuals, there were 12 men and 6 women aged between 59 and 87 years (75.8±7.9 years). All included participants had experienced tremors for at least 11 months, with the majority experiencing a moderate severity of hand tremor (a score of 2.11±0.90 on the Fahn–Tolosa–Marín Tremor Rating Scale, FTM). The demographic and clinical characteristics of the individuals are detailed in [13].

### 2.3. Experimental Procedure for Data Collection

During the experimental procedure, participants sat comfortably in a chair with their feet flat on the floor and their backs against the backrest. They were instructed to maintain the posture of the most affected upper limb, that is, arms parallel to the transverse plane and 90º from the coronal plane. Thus, the postural tremor was evaluated with the elbow joint extended and the forearm pronated, as shown in Figure 3. In order to minimise the influence of the movement of other parts of the body and avoid limb fatigue, an arm rest supported the arms of the subjects during the experiments.

All individuals were assessed with and without peripheral mechanical stimulation. The most affected upper limbs (hand and forearm) of participants were subjected to five vibrotactile stimulation paradigms. Each stimulation paradigm lasted 240 s, and a 10 min rest period was considered between them, as illustrated in Figure 4. Thus, each participant completed the experiment in about 60 min.

Initially, postural tremor was recorded in the absence of vibrotactile stimulation for four minutes in order to characterise involuntary activity without the influence of mechanical inputs. This vibrotactile stimulation paradigm (called baseline) corresponded to a control measurement for making comparisons, thus characterising the tremor of each individual during upper limb posture maintenance.

Subsequently, the participants were subjected to four vibrotactile stimulation patterns, which included both constant frequencies (50 and 250 Hz) and alternating frequencies (with an increasing frequency between 50 and 450 Hz with a resolution of 50 Hz and these frequencies in random order). Thus, the limbs were stimulated at 50 Hz, then 250 Hz, increasing the linear frequency of stimulation, and finally randomly, in the order depicted in Figure 4.

In addition, each of these four stimulation patterns lasted approximately four minutes. These stimulation patterns were subdivided into a pre-stimulation period (1 min), a stimulation period (2 min), and a post-stimulation period (1 min), as illustrated in Figure 4. This subdivision was considered in order to evaluate postural tremor before, during, and after the application of the vibratory stimulation.

A custom wearable device, previously detailed in [13], was employed to mechanically stimulate the upper limb and quantitatively measure involuntary activity before, during, and after stimulation, as depicted in Figure 3. Vibrotactile stimulation was applied to the fingertips, palm of the hand, and forearm using piezoelectric actuators. To capture tremor, triaxial angular velocities of the forearm and hand were collected with a pair of inertial measurement units.

### 2.4. Signal Processing

Data analysis and signal processing were performed using R (version 4.2.3) [25], a programming language and an open source for data visualisation and statistical analysis. In this study, a customised R package was designed to support tremor analysis and perform biomedical signal processing. The programs were developed in RStudio (version 2022.12.0) [26] integrated development environment. Figure 5 presents the steps used to preprocess signals for feature extraction and statistical analysis.

#### 2.4.1. Signal Visualisation

In this study, each collected signal was carefully visualised. For each participant, 30 signals were visually inspected (5 stimulation patterns × 2 positioning regions of IMUs × 3 axes). Figure 6 depicts gyroscope signals captured from an individual while maintaining the posture of the most affected limb for various stimulation paradigms. Thus, for 18 participants, the data set consisted of 540 signals.

This step is frequently overlooked or not mentioned in many signal processing studies. However, it is an important step in the analysis and processing of any biomedical signal. It is significant for several reasons. Signal visualisation improves the understanding of the analysed phenomenon by displaying the behaviour and trends in the signal and guiding signal processing and statistical analyses. Furthermore, this step avoids the use of unnecessary complex tools and allows for data integrity verification.

#### 2.4.2. Signal Preprocessing

The visualisation of the signals revealed the presence of trends as well as artefacts produced by the vibratory stimulus. In this sense, wavelet decomposition was employed to eliminate trends and stimulus-related noise from the signals of each body segment, obtaining only the tremor-related information for the hand and forearm. The signal was decomposed into five components using the Daubechies orthonormal compactly supported wavelet of Length 8.

The components resulting from signal decomposition include details (d1, d2, d3, d4, and d5) and approximation (s5). The approximation component detects the low-frequency oscillations of the signal, whereas the detail components detect the high-frequency content of the signal. The strategy for signal filtering was to identify the wavelet component that best represented the noise activity and then reconstruct the signal based on the algebraic summation of its components, excluding the component associated with noise and trends, as illustrated in Figure 7.

##### Trend Removal

Generally, inertial signals are subjected to linear and non-linear trends [27]. These trends are systematic changes that occur in the signal over time. Basically, they originate from sources such as the influence of the acceleration of gravity and the Earth’s magnetic field, small motions unrelated to the motion of the investigated limb, or even the signal acquisition process itself (e.g., cable movement) [28].

In such cases, one of the most important preprocessing steps is to identify, model, and remove the trends to minimise the influence of offsets and fluctuations in the signals that do not refer to the phenomenon investigated. Because linear and nonlinear trends are associated with low-frequency oscillations, the strategy for removing this type of noise from the signals was to reconstruct the signal by removing the approximation component that best captured the signal trend.

Figure 8 shows an example of the outcome of the trend removal using wavelets. The original signal (top) was influenced by both linear and nonlinear trends, as shown in the figure. Because the s5 component, which is the slowest approximation component, represents the signal trends, it was necessary to reconstruct the original signal by adding up all detail components except the s5 component.

##### Noise Removal

In addition to trends, inertial signals may be susceptible to artefacts that introduce seasonal fluctuations or more complex components into the signals. This is because the inertial signals can be corrupted by external inputs, such as peripheral mechanical stimuli, that cause the investigated system to oscillate during signal acquisition. It was verified that signal component d1 captured this type of noise. Thus, the signal was filtered by reconstructing the signal without d1. Figure 9 shows an example of the outcome of noise removal using wavelets.

#### 2.4.3. Time Domain Analysis

##### Selection, Classification and Windowing of Signal Regions

In order to compare the dynamics of postural tremor before, during, and after each stimulation, the signals were segmented into three regions. For this purpose, the controlling signal of the piezoelectric actuator (Figure 10), utilised during each stimulation, was employed to divide the signal into pre-stimulus, stimulus, and post-stimulus regions. In the baseline paradigm, where no stimulation occurred, the signal was not segmented, and the entire signal was considered for signal windowing.

The signal windowing stage is depicted in Figure 11 for the vibrotactile stimulation case. To window the signal from each region, a nonoverlapping rectangular window of 500 ms was employed. The purpose of this stage was to segment the signal such that a set of features from each region could be estimated. A similar procedure was adopted for the windowing of baseline signals.

##### Feature Extraction

The set of estimated features from each window was summarised by the median of the feature. Two features were investigated, i.e., Approximate Entropy (ApEn) and Mean Absolute Value (MAV).

Approximate Entropy (ApEn) is a feature that expresses the regularity and predictability of the signal over time. This feature is defined as
(1)$$ApEn\left(m,r,N\right)=\left\{\begin{array}{cc} \phi^{m}\left(r\right)-\phi^{m+1}\left(r\right), & \forall \;m\leq 0\\ -\phi^{1}\left(r\right), & m=0 \end{array}\right.,$$
where
(2)$$\phi^{m}\left(r\right)=\frac{1}{N-m+1}\sum_{i=1}^{N-m+1}ln\left(C_{i}^{m}\left(r\right)\right),$$
*N* corresponds to the length of the signal, $m\in Z^{+}$ corresponds to the embedded dimension, which represents the length of two segments in a sequence to be compared, $r\in R^{+}$ corresponds to the filter factor, which shows the tolerance for accepting similar patterns between two segments and
(3)$$C_{i}^{m}\left(r\right)=\frac{N_{i}^{r}}{N-m+1}$$
with $N_{i}^{r}$ representing the number of Euclidean distances between each the two segments (*m*—dimensional delay vectors), which is no larger than *r*.

This definition implies that the approximate entropy is a value between zero and two. When ApEn tends to zero (ApEn→0), the signal is more regular and deterministic over time. This implies that the presence of repeating patterns in the signal makes it more predictable and thus less complex. If ApEn tends to two (ApEn→2), then the signal is more unpredictable, that is, more complex over time.

Mean Absolute Value (MAV) is a feature that captures changes in the amplitude of the signal. It is calculated by averaging the absolute values, that is,
(4)$$MAV=\frac{1}{N}\sum_{i=1}^{N}\left|x_{i}\right|,$$
where *N* corresponds to the number of samples of the signal and $x_{i}$ corresponds to the *i*th signal sample. Therefore, high MAV values imply a high order of magnitude of the signal and, consequently, of the measured quantity.

In the analysis of tremor, the assessment of amplitude and regularity may be very important to characterise the involuntary activity. This is because tremor amplitude describes the severity of involuntary movements in the affected limb, while tremor regularity allows identifying motor fluctuations and differentiating tremors or periods of involuntary activity even when they present similar amplitudes over time.

#### 2.4.4. Frequency Domain Analysis

##### Power Spectrum Estimate

To analyse the frequency content of a tremor and provide information about its spectral characteristics, the Power Spectral Density (PSD) was estimated for each signal, employing the adaptive sine-multitaper approach described by Barbour and Parker [29]. The PSD of a signal describes the power distribution as a function of frequency. Therefore, it is a method that provides insight into the predominant frequencies and their relative power levels.

##### Feature Extraction

For the feature extraction step, the PSD was smoothed (a Savitzky–Golay filter of order 7 and length 19) to ease the estimation of features. Two features were estimated from the PSD: peak power ($p_{peak}$) and peak frequency ($f_{peak}$). Peak power corresponds to the global maximum value of the PSD, while peak frequency is the frequency associated with this power value. In the context of tremor analysis, these features are of utmost importance as they describe the component of the highest power, i.e., the main component of the signal.

### 2.5. Correlation Analysis

In order to determine whether there was any relationship between the objective assessment of tremor proposed in this study and the clinical assessment, Spearman’s correlation coefficients were estimated for the set of features extracted from the signals without vibrotactile stimulation. The strength of the correlation was considered to be very weak (ρ∈]0.00;0.20[), weak (ρ∈[0.20;0.40[), moderate (ρ∈[0.40;0.60[), strong (ρ∈[0.60;0.80[) and very strong (ρ∈[0.80;1.00]) [27], considering a significance level of 0.05.

### 2.6. Regression Analysis

To further demonstrate the relevance of the features and to better understand the influence of variability on the relationship among variables, a regression analysis based on the augmented dataset was carried out. The possibility of establishing a relationship between variables is a good indicator of the quality of the experimental samples used in the study.

#### 2.6.1. Data Augmentation

The augmentation of the dataset was performed by a generative model based on a multivariate Gaussian distribution, using the R package MASS [30]. For each of the four clinical score levels, the mean and covariance matrix of the feature set given by two-dimensional feature vectors (peak power and frequency) were estimated to represent the distribution of the initial data and generate new data points by sampling from a multivariate Gaussian distribution, preserving the relative percentage of different clinical score levels.

#### 2.6.2. Data Modelling

For data modelling, the augmented data were randomly partitioned into an 80% training set and a 20% test set. Then, a Feed-Forward Neural Network algorithm was used to estimate the regression model, given by the model in Equation (Equation 5). The use of a neural network does not make any assumptions about the underlying distribution of the data and captures linear and nonlinear relationships among variables. The neural network had three layers, i.e., input (two neurons), hidden (10 neurons), and output (one neuron). The R package nnet was used for training the neural network [30].
(5)$$Clinical\;\;score∼peak\;\;power+\frac{1}{peak\;\;frequency}.$$

After training, the model was applied to the test set. In addition, the model’s performance on the test set was evaluated using the mean squared error (MSE).

### 2.7. Statistical Analysis

An overview of the statistical analysis is presented in Figure 12. For each feature vector,
(6)$$F_{k}=\left\{f_{1}^{k},\cdots ,f_{N}^{k}\right\},$$
where $k\in \{MAV,ApEn,p_{peak},f_{peak}\}$, N∈{1,⋯,18} corresponds to the number of participants and $f_{i}^{k}$ corresponds to the *k*-th feature of the *i*-th participant, the Bootstrap method with a 95% confidence interval was used to estimate the distribution of the medians given the axis (A={X,Y,Z}), the position of the IMU (P={*dorsum of the hand (1), forearm (2)*}), the stimulation paradigms (W={*baseline, 50 Hz, 250 Hz, linear increase, random*}), and, specifically for the time domain analysis, the type of activity region (R={*pre-stimulus, stimulus, post-stimulus*}). The estimation of the medians of each $F_{k}$ considered *B* = 10,000 bootstrap samples.

For the time domain, the statistical analysis considered a linear model for the evaluation of the relationship between the feature value ($S_{MAV}$ and $S_{ApEn}$) and the type of region of activity, taking into account the axis, IMU positioning, and stimulation paradigm. These models are defined in Equation (Equation 7).
(7)$$S_{k}\;|\;A,P,W∼R.$$

For the frequency domain, the model of Equation (Equation 8) was considered to evaluate the relationship between the feature value ($S_{p_{peak}}$ and $S_{f_{peak}}$) and the stimulation paradigm type, given an axis and IMU positioning.
(8)$$S_{k}\;|\;A,P∼W.$$

Initially, parametric analysis (one-way ANOVA) was attempted, and once the assumptions of such an approach were not verified, nonparametric analysis (Kruskal–Wallis test) was carried out, considering a *p*-value < 0.05. Thus, the normality assumption of one-way ANOVA was verified by the Kolmogorov–Smirnov test (*p*-value > 0.05), and Levene’s test verified the homogeneity of variances (*p*-value < 0.05).

For the time domain, the statistical analyses were delineated to evaluate statistical differences among the periods of activity (period before stimulation, period during stimulation, and period after stimulation) for the bootstrap variables ($S_{MAV}$ and $S_{ApEn}$), taking into account each axis, positioning of the IMU, and stimulation paradigm. For the frequency domain, the statistical analyses were delineated to evaluate differences among the stimulation paradigms (baseline, 50 Hz, 250 Hz, linear increase, and random) for the bootstrap variables ($S_{p_{peak}}$ and $S_{f_{peak}}$), given each axis and positioning of the IMU.

In addition, the effect size associated with the adopted statistical test was employed to verify the magnitude of the difference between groups for the analyses in the time and frequency domains. This statistical metric provides a quantitative interpretation of the difference. Basically, the larger the effect size, the greater the difference between groups.

After verifying the statistically significant difference, a post hoc analysis was performed to determine the levels at which there was a statistically significant difference between the regions of activity (time domain analysis) and stimulation paradigms (frequency domain analysis). Multiple pairwise comparisons were performed by means of the Tukey Honest Significant Differences in the case of the parametric approach or the pairwise Wilcoxon rank-sum test in the case of the nonparametric approach using the Bonferroni correction.

## 3. Results

### 3.1. Correlation Analysis

Results of correlation analysis between clinical and instrumental assessment are shown in Figure 13. The features sensitive to changes in tremor amplitude, whether in the time (MAV) or frequency (peak power) domains, showed a significant correlation with clinical rating scale (*p*-value <0.05). Regardless of the sensor axis and positioning, the strength of these relationships was very strong (ρ≥0.81) for the MAV features estimated from the baseline signals, and strong (ρ≥0.77) for the peak power.

The peak frequencies also exhibited strong associations with the clinical scores. However, these correlations were negative (−0.68≤ρ≤−0.66), and specifically for the forearm, they were significant only for the X axis. Additionally, the clinical scores positively correlated only with the regularity values of the X axis of the sensor positioned on the hand, but the correlation was moderate (ρ=0.48, *p*-value <0.05).

### 3.2. Regression Analysis

Using the augmentation technique, 10,100 samples were generated based on the initial distribution of the data (18 two-dimensional samples extracted from the baseline signal around Y axis). Figure 14 contrasts the augmented data with the initial data. Despite increased variability and overlapping levels in the augmented data, discernible patterns still exist that characterise each profile of the different clinical scores.

Most of the predicted scores followed the expected output (Figure 15). Although some estimated scores not following the expected output are due to error estimates, the low value of the MSE (0.09) suggests that the model exhibits good performance, which corroborates the results shown in Figure 15.

### 3.3. Time Domain Analysis

The boxplots depicted in Figure 16 and Figure 17 show the distribution of amplitude (MAV) and entropy (ApEn) for four activity regions (baseline, pre-stimulus, stimulus, and post-stimulus), considering each stimulation pattern (50 Hz, 250 Hz, linear increase, and random), axis (X, Y, and Z) and sensor placement (hand and forearm).

#### 3.3.1. Amplitude Analysis

In general, regardless of the sensor’s positioning and axis, the amplitude of involuntary activity without stimulation was higher than the amplitude during the pre-stimulus for all stimulation patterns (Figure 16 and Figure 17). However, when stimulating the hand at 50 Hz, no statistically significant difference in amplitude around the Z axis between these two regions was observed ($\eta_{H}^{2}=0.016$, *W* = 50,880,056, *p*-value =0.18).

The amplitudes during the pre-stimulation and stimulation periods exhibited statistically significant differences (*p*-value <0.05). The amplitude of involuntary activity in the stimulation period was greater than the amplitude of the pre-stimulus region for the 50 Hz, increasing linear frequency, and random frequency stimulation patterns, as shown in Figure 16 and Figure 17.

Similarly, when the forearm was stimulated at 250 Hz, the amplitude of involuntary activity in the stimulation period was greater than the amplitude in the pre-stimulus period ($\eta_{H}^{2}\geq 0.125$), with the exception of the Y axis of the sensor in the forearm ($\eta_{H}^{2}=0.033$). However, an opposite behaviour was observed when the hand was stimulated at 250 Hz, which caused a reduction in the amplitude ($\eta_{H}^{2}\geq 0.101$).

The amplitude of the upper limb involuntary activity was higher in the post-stimulus than in the pre-stimulus period for the 50 Hz, increasing linear frequency, and random frequency stimulation patterns (Figure 16 and Figure 17). Regardless of the sensor’s position and axis, comparisons between the amplitudes of these two regions revealed statistically significant differences ($\eta_{H}^{2}\geq 0.016$, *p*-value < 0.05).

On the other hand, the hand and forearm exhibited distinct responses to stimulation when stimulated at 250 Hz (*p*-value < 0.05). This stimulus led to a decrease in amplitude in the hand and an increase in the forearm (Figure 16 and Figure 17), except for the Y axis of the sensor in the forearm, which best represented the involuntary movements of the majority of participants.

#### 3.3.2. Regularity Analysis

In terms of regularity, a random behaviour of entropy was observed when comparing the baseline with the pre-stimulus period of stimulation patterns. There was no statistically significant difference between the entropy of these two regions for the X ($\eta_{H}^{2}=0.175$, *W* = 46,982,996, *p*-value = 0.18) and Z ($\eta_{H}^{2}=0.012$, *W* = 47,992,388, *p*-value = 0.06) axes when the hand was stimulated with an increasing frequency and random frequency, respectively.

When comparing the pre-stimulus and stimulus periods, the entropy values exhibited considerable variability among the different axes and stimulus patterns, sometimes decreasing and sometimes increasing (*p*-value < 0.05). Despite the overall increase in amplitude observed for most stimulation patterns during the stimulation phase, the effect of stimulation on the regularity of a limb’s involuntary activity varied, sometimes resulting in an increase in regularity and sometimes in a decrease.

Among the stimulation patterns, the 250 Hz stimulus altered the dynamics of the tremor after stimulation, making it more complex despite the reduction in amplitude, particularly noticeable in the involuntary hand activity. In fact, the entropy of involuntary activity in the hand and forearm during the pre-stimulus was lower than the entropy in the post-stimulus at 250 Hz, except for the Y axis of the hand-positioned sensor, which showed no statistical difference ($\eta_{H}^{2}=0.012$, *W* = 48,729,552, *p*-value = 1.00).

Increasing the frequency of stimulation also resulted in a similar response in the hand. This stimulus led to decreased entropy of involuntary activity during the pre-stimulus period compared to the post-stimulus period ($\eta_{H}^{2}\geq 0.022$, *p*-value < 0.05), except for the Y axis ($\eta_{H}^{2}=0.251$, *p*-value < 0.05). The differences between the entropies before and after stimulation varied for the other stimuli, but all consistently demonstrated statistically significant differences.

### 3.4. Spectral Analysis

Both for each individual (a qualitative analysis) and for the group of participants (a quantitative analysis), spectral analysis of tremor was performed to characterise postural tremor and to determine whether different stimulation frequency patterns could alter tremor characteristics, such as power and frequency.

#### 3.4.1. Qualitative Analysis of Postural Tremor

Figure 18, Figure 19 and Figure 20 provide an overview of the spectral analysis for three participants (Individuals 1, 7 and 17). Each figure depicts the power spectral densities of the angular velocities of the hand and forearm for the participants during posture maintenance for both the trial without vibrotactile stimulation and the trial with vibrotactile stimulation. In addition, the figures highlight the spectral behaviour of the frequency bands adjacent to the peak frequency.

In general, the involuntary movement of the hand and forearm was caused by the superposition of multiple components with different frequencies and amplitudes for each individual during posture maintenance in the absence of vibrotactile stimulation. For the majority of the participants, spectral distributions were bimodal or multimodal, with distinct peaks in various frequency bands, as exemplified in Figure 18, Figure 19 and Figure 20. For a more comprehensive insight into the spectral behaviour of the other participants, see Appendix A.

Spectra with multimodal distributions exhibited three or more predominant peaks, but of unequal magnitudes, e.g., the spectrum of Individual 1 (Figure 18). In these cases, the spectra contained a major peak (i.e., a global maximum) and minor peaks (i.e., local maxima) in the bands adjacent to the peak frequency. For the majority of participants, the peak frequencies of the involuntary activity of the upper limb ranged from 4 Hz to about 9 Hz. Despite the low levels of magnitude on the spectra, the lower side-band exhibited peaks at 1–6 Hz, while the upper side-band showed peaks between 7 and 15 Hz.

Similarly, the spectra with bimodal distributions had two predominant peaks of different magnitudes, e.g., the spectra of Individuals 7 and 17 (Figure 19 and Figure 20). All spectra with this distribution had a major and minor peak. Nevertheless, for some spectra, the minor peak was present on the lower frequency side of the dominant peak (1–5 Hz), as seen in the spectrum of Individual 17 (Figure 20). For others, the minor peak was present on the higher frequency side of the dominant peak (7–15 Hz), as seen in the spectrum of Individual 7 (Figure 19).

For some spectra with bimodal distribution, the frequency of the involuntary activity ranged between 3 and 5 Hz. In these cases, the spectra exhibited a minor peak in the upper side-band (7–15 Hz). In other spectra with bimodal distribution, a dominant peak was presented around 7–9 Hz, but the spectra also contained a minor peak at 1–5 Hz (i.e., at lower frequencies).

Regardless of the type of spectral distribution, all participants exhibited an 8–12 Hz component in their involuntary activity. The magnitude of this component varied among the participants. For some of them, the 8–12 Hz component had the highest magnitude, despite the presence of low-frequency components (4–7 Hz) more characteristic of ET, as illustrated in Figure 20. For others, this component was not significant, having a very low magnitude in relation to the component with the major peak (Figure 18).

In a body segment, the tremor can have multiple components, and the multiple peaks in the spectra suggest the coexistence of tremor components in distinct frequency bands for the individuals with ET. As observed in the spectra of Individuals 1, 7, and 17 (Figure 18, Figure 19 and Figure 20), a vibratory input can modify one of these components to a greater or lesser extent, exhibiting contrasting effects or no impact on the main component.

In summary, the effect of vibrotactile stimulation on the main component of upper limb involuntary activity varied among individuals, as depicted in Figure 21. For some participants, the power of the involuntary activity decreased with the different patterns of vibrotactile stimulation. For others, the response was the opposite, leading to an increase in power. Furthermore, only a few individuals did not show a pronounced change in the magnitude of the involuntary activity during stimulation of the upper limb (less than 5% difference).

Among the participants, two individuals (Individuals 4 and 8), both with mild tremors (a score of two on the clinical scale), experienced an increase in the maximum power of the involuntary activity for all stimulation patterns. On the other hand, there were three individuals (Individuals 1, 6 and 9) with moderate tremors (a score of 2.67 ± 0.58 on the clinical scale); all stimulation patterns provided a reduction in the maximum power of the involuntary hand and forearm activity, as shown in Figure 21.

For many individuals, the stimuli with a constant frequency of 250 Hz and random frequency played an important role as short-term tremor suppressors. More than half of the participants presented a decrease in the maximum power when the hand and forearm were stimulated with these frequency patterns, as shown in Figure 22. On the other hand, the stimulus at 50 Hz was the stimulation pattern that most induced the increase in power (Figure 22). The increased power when the limb is stimulated at 50 Hz suggests that this stimulus may interfere with some processes that triggers postural tremor for some individuals with ET.

#### 3.4.2. Quantitative Analysis of Postural Tremor

The boxplots of the distribution of the peak frequency and peak power are shown in Figure 23 and Figure 24, respectively, for each axis and sensor position. A shift in the powers and frequencies occurred for different stimulation patterns when compared to the condition without vibrotactile stimulus.

When there was no stimulation, the peak power was higher on the hand than on the forearm, regardless of sensor axis, as depicted in Figure 23. Although involuntary movement is a three-dimensional complex phenomenon, sensor axes Y (suggesting a tremor characterised by flexion and extension movements) and X (suggesting a tremor characterised by pronation and supination movements) on the hand (IMU 1) and forearm (IMU 2) were more sensitive to detect it. These two axes resulted in a higher power peak and greater variability, as seen in Figure 23.

The 250 Hz and random frequency stimulation patterns reduced the power for most axes in the hand and forearm, except for the Y axis, which was positioned at the forearm. The 50 Hz stimulation pattern increased the power of involuntary activity on all axes except the Z axis of the IMU at hand. These findings corroborate the analysis conducted for each participant (Figure 21 and Figure 22).

The peak frequency evaluation revealed that, for the hand, there was usually a shift of the peak frequency to lower frequencies, whereas for the forearm, the effect was generally the opposite, with a shift of the peak frequency to higher frequencies. Furthermore, a common response pattern for the peak frequency behaviour was observed in the sense that there was an alternate response in the frequency as a function of the stimulus pattern. For example, a decrease from the baseline to 50 Hz, an increase from 50 Hz to 250 Hz, a decrease from 250 Hz to the linear increasing frequency stimulus, and finally an increase from linear to random stimulus were observed.

Interestingly, the same alternating pattern was observed for power, but in the opposite direction when compared to frequency. For instance, there was an increase from the baseline to 50 Hz, a decrease from 50 Hz to 250 Hz, an increase from 250 Hz to the linear increasing frequency stimulus, and a decrease from linear to random. In summary, when the frequency was shifted to lower frequencies, the power increased, and when the frequency was shifted to higher frequencies, the power decreased.

For the peak frequencies and powers, significant differences were found between the scenarios with and without stimulation for all axes and sensor positions ($\eta_{H}^{2}\geq 0.107$, *p*-value < 0.05). Multiple pairwise comparison between groups, given by the Wilcoxon rank-sum test with Bonferroni correction, suggested statistically significant differences between most pairs, as presented in Figure 23 and Figure 24.

When comparing peak powers, there was no statistically significant difference (*p*-value = 1.00) observed in the powers of the X axis in the forearm when it was not stimulated compared to when it was stimulated with increasing linear frequency. Additionally, no significant differences (*p*-value = 1.00) were found in the peak frequencies of involuntary hand activity around the Y and Z axes between the stimulus with a constant frequency of 250 Hz and the scenario without stimulation, nor between the stimulus with a random frequency and the scenario without stimulation, respectively.

## 4. Discussion

### 4.1. Relationship between Clinical and Instrumental Assessment

The positive and strong correlations observed between features sensitive to changes in tremor amplitude and scores on the clinical scale (Figure 13) suggest that, like the clinical scale, the estimated measurements of signals captured by sensors have the ability to accurately depict the severity of motor signs in terms of amplitude. Consequently, there is a corresponding increase or decrease in sensor measurements as the ratings for clinical tremor severity progress.

As inertial sensors (e.g., gyroscopes) capture subtle movements that are often undetectable to the naked eye, a strong correlation indicates that the examiner also detects and distinguishes low-amplitude involuntary movements. Therefore, these strong correlations with the clinical evaluation indicate effective administration of the clinical instrument and good clinical practice.

On the other hand, the negative and strong correlations observed between peak frequencies and clinical scores imply an inverse relationship between the clinical severity of the tremor (in terms of amplitude) and its dominant frequency (Figure 13). This implies that tremors classified as more clinically severe (with greater amplitude) tend to have lower frequencies, whereas tremors of lesser magnitude typically exhibit higher frequencies.

The absence of overall correlations between clinical scores and entropy values might result from the clinical scale’s inherent limitation in measuring the temporal regularity of tremor. Although the clinical scale is an important tool for determining the status of ET, it has inherent limitations and does not allow for a clear assessment of the frequency of oscillation of the affected limb, as well as changes in the amplitude, regularity, or rhythmicity of the tremor over time, which are characteristics that are difficult to assess only visually but which are important for characterising ET and, more than that, understanding tremor.

In addition, the proposed regression model has demonstrated significant potential for accurately assessing the severity of tremor based on the spectral features (peak power and peak frequency) extracted from inertial signals (MSE of 0.09). This model could be embedded in wearable inertial devices to monitor tremor severity in real time in response to a vibratory input or, more than that, other types of external input.

### 4.2. Characterisation of Involuntary Activity in the Absence of Vibrotactile Stimulation

Spectral analyses conducted for each participant suggested that the postural tremor in the hand and forearm of individuals was caused by the superposition of multiple components with different frequencies and energies. The participants exhibited bimodal or multimodal spectral distributions with distinct peaks in different frequency bands, as seen in the PSDs of the involuntary activities of the hand and forearm without stimulation of three participants (Figure 18, Figure 19 and Figure 20).

In this study, certain participants showed a peak frequency associated with higher power in the range of approximately 3.5 to 5 Hz, while others demonstrated a peak frequency ranging from about 6 to 9 Hz. The tremors exhibited by them fall within the expected frequency range, as individuals with ET typically exhibit tremor frequencies ranging from 4 to 12 Hz [8].

However, this variation among participants may be attributed to differences in age. This is because it is believed that the frequency of the tremor reduces over the years [19,31]. Another hypothesis that could explain this difference is the marked presence of physiological tremor, which is characterised by a higher frequency band in the 8–12 Hz range [10], which would contribute more to involuntary activity during posture maintenance than the components at 3–6 Hz.

The lower energy peaks of both bimodal and multimodal distributions were typically located in frequency bands adjacent to the peak frequency. Spectra that had lower magnitude peaks in the upper side-band often exhibited peaks in the 7–15 Hz range, while spectra that had lower magnitude peaks in the lower side-band typically contained peaks in the 1–6 Hz range.

Interestingly, the presence of these lower power components in the spectra has also been reported, to a greater or lesser extent, in electrophysiological studies investigating the possible origins and underlying mechanisms of tremor, such as the recent study by Vial and colleagues [32] and the pioneering studies in this area [10,19,33], which dramatically widened the field of tremor in terms of frequency characterisation.

In this context, the minor peaks with frequency between 1 and 3 Hz raise the hypothesis that the components in this frequency band are associated with ataxic tremors, which are involuntary movements often associated with cerebellar pathologies or the response to reflexes activated by peripheral mechanisms, and are not considered to be a tremor in themselves but rather a dysmetria [33]. The presence of these components in some individuals may be related to instabilities in the reflex pathways during posture maintenance. This is because optimal motor control of posture results from the combination of peripheral information with peripheral feedback from muscle spindles, joint receptors, and mechanoreceptors [34].

On the other hand, the minor peaks with a frequency between 4 and 6 Hz raise the hypothesis that the components in this frequency band are associated with pathological tremors [33]. Therefore, the presence of components in this frequency band suggests the existence of typical oscillations present in diseases such as ET, which is characterised by the presence of tremors during posture maintenance.

Finally, the minor peaks in the higher frequency band, typically ranging between 8 and 12 Hz, give rise to the hypothesis that these components are associated with natural physiological processes characteristic of physiological tremor [10,33]. This type of tremor is not commonly seen by the naked eye; however, several factors can make it noticeable, such as excessive physical exertion, muscle fatigue, or emotional stress [16].

Regardless of the type of spectral distribution, the involuntary activity of all participants showed an 8–12 Hz component, which can be associated with physiological tremor. This, in turn, suggests the presence of physiological tremor to varying degrees during posture maintenance. Furthermore, it implies that this component may be one of the components that contribute to the severity of involuntary activity during limb posture, as a constructive summation of the ET components with the physiological tremor components.

### 4.3. Mechanical Vibrations: A Way to Change Tremor Dynamics?

Tremor is a complex phenomenon. It probably arises in circuits responsible for the control and coordination of body movements and, consequently, combines central and peripheral mechanisms [10,16]. However, the interaction of these circuits for the generation and manifestation of tremor in ET still remains poorly understood, which motivates the investigation of tremor and the effect of external inputs on its mechanisms.

In this sense, several studies have been outlined in recent years in order to understand the effect of external inputs on tremor and investigate possible ways to change its behaviour through the application of inertial loads and electrical stimulation [5,6,10,12,14,16,19,20,21]. Although there is limited evidence regarding the effect of mechanical stimulation on tremors, the idea of employing vibrations is not a recent one. During the 19th century, the work of the neurologist Jean-Martin Charcot popularised the idea of employing vibrations to treat tremor [35]. Charcot proposed a vibratory chair to relieve the symptoms of Parkinson’s disease, reporting successful outcomes among his patients after a few sessions [36].

The use of vibrations in the treatment of Parkinson’s disease stemmed from the observation that individuals with this condition exhibited a decrease in resting tremor when travelling in carriages or on railway tracks [35,36]. However, few studies have followed up with investigating the feasibility of vibration in Parkinson’s disease or other tremor disorders after the death of Charcot in 1893 [35]. This may be justified by the fact that the possible physiological mechanisms behind vibrations, as well as their relationship to the circuits responsible for the generation and manifestation of tremor, were unknown at the time, although empirical evidence suggested a potential positive effect of vibrations.

Thus, the motivation for this research was to investigate the effect of distinct vibrotactile stimulation patterns on the postural tremor of individuals with ET. As the vibratory pathway (Figure 2) shares connections with the circuits underlying the generation and manifestation of tremor in the ET (Figure 1), this study hypothesised that the introduction of perturbations in the afferent pathways may play an important role in the dynamics of tremor.

In this study, the involuntary movements of 18 participants with ET were evaluated by utilising inertial sensors placed on the hand and forearm of the limb most affected by the condition. This assessment was conducted both in the absence of vibrotactile stimulation and while exposed to four distinct vibratory stimuli, each with different frequencies: 50 Hz, 250 Hz, increasing linear frequency, and random frequency (Figure 4).

The study verified that peripheral vibrotactile stimulation was able to alter the dynamics of tremor in individuals with ET, changing tremor characteristics like amplitude, peak power, peak frequency, and regularity over time. However, the findings suggest that the motor response to different stimulation patterns may be variable among individuals. This variability may be due to various reasons, such as the specificity and size of the receptive field of vibration-sensitive receptors, as well as the intensity and duration of the stimulus applied [37,38].

The findings revealed that, overall, the 50 Hz, increasing linear frequency, and random frequency stimuli caused an increase in the amplitude of involuntary activity in both the hand (Figure 16) and forearm (Figure 17) during and after stimulation compared to the pre-stimulus period. On the one hand, this increase can be explained by the tremor superimposing on the mechanical input, as if their oscillations experienced constructive interference, resulting in an oscillation corresponding to the algebraic summation of each individual oscillation. On the other hand, this increase may be linked to the stimulation patterns amplifying tremor components.

Nonetheless, this increase in amplitude resulting from these three stimulation patterns was accompanied by varying entropy behaviour. This finding is intriguing because it suggests that while these patterns increased the tremor amplitude, some patterns made it more regular over time, resulting in a more pronounced tremor but with fewer motor fluctuations. In contrast, other patterns led to a more severe tremor with increased motor fluctuations. This result is significant because, even though these stimulation patterns do not suppress or reduce the tremor, they enable the characterisation of a more severe tremor with or without motor fluctuations over time.

In particular, these stimulation patterns, which induce an increase in tremor amplitude and alter its regularity, have the potential to serve as an individual reference for assessing tremor severity, for example, using vibratory stimuli to set a standard for severe tremor for each patient with ET. This is significant because there is an inherent variation in the values designated by current clinical rating scales, such as the Fahn–Tolosa–Marín Tremor Rating Scale (FTM) [39] and the Tremor Research Group Essential Tremor Rating Scale (TETRAS) [40], for mild, moderate, and severe tremor.

Although these clinical scales assess tremor amplitude similarly through visual inspection, they assign different scores to different amplitude ranges. This variation can lead to different classifications when a patient’s tremor amplitude falls within overlapping ranges of the scales or at the borderline between two scores. Consequently, this variability between the different clinical scales can lead to even greater heterogeneity among patients, as individuals with the same tremor severity can be classified with different levels of severity. This, therefore, demonstrates the difficulty clinical scales have in consistently assessing quantitative aspects of tremor (e.g., amplitude), although they are currently important tools for determining the severity of ET.

In contrast to the responses observed with 50 Hz, increasing linear frequency, and random frequency stimuli, the amplitude of involuntary activity during and after 250 Hz stimulation was lower than the amplitude recorded in the pre-stimulation period. This reduction was only absent on the X and Z axes of the sensor placed on the forearm. Despite the reduction in amplitude for the Y axis of the sensor in the forearm, which was the axis that best represented the involuntary movements of most participants, the different responses between the hand and the forearm suggest that different body segments can respond differently to the same pattern of stimulation.

These findings may be associated with the fact that there is maximum sensitivity in the fingertips when a vibratory stimulus at 250 Hz is applied to this region [41]. Specifically, the sensitivity at this frequency (250 Hz) can be elucidated by the significant role of Pacinian corpuscles, which exhibit pronounced sensitivity to vibrations in the high-frequency range (60–400 Hz) [42] and optimal sensitivity precisely at 250 Hz [43]. Although these mechanoreceptors are present in different regions of the glabrous skin (hairless skin), they are more abundant on the fingertips, which was one of the main regions stimulated in this study (Figure 3) compared to the glabrous skin of the forearm or palm of the hand [44].

Additionally, there was an increase in the entropy of both the hand and the forearm during and after stimulation at 250 Hz. This indicates a change in the type of tremor from a more predictable signal, similar to a sinusoidal pattern, to a non-predictable signal. Therefore, stimulation at 250 Hz, especially of the hand, led to motor fluctuations with low amplitude. This suggests that the 250 Hz stimulus has the potential to be incorporated into the development of technologies for tremor reduction. For example, a smartwatch could be designed to monitor hand tremor and apply 250 Hz stimulation patterns to reduce tremor.

The results of the spectral analysis further reinforce the findings of time domain analysis. For many participants, the 250 Hz stimulus resulted in a decrease in peak power when compared to the peak power estimated from involuntary activity in the absence of stimulation, as depicted in Figure 21 and Figure 22. Although a decrease in peak power was also observed with the random frequency stimulus (Figure 21 and Figure 22), the 250 Hz stimulus generally exhibited less variation among participants, regardless of axis or sensor placement (Figure 23). Interestingly, the random frequency stimulus that showed reduction in tremor also had a frequency of 250 Hz.

Moreover, in comparison to the spectrum of involuntary activity in the absence of vibrotactile stimulation, the stimulation at 250 Hz led to a shift in peak frequency (Figure 24), with the exception of the Y axis of the sensor positioned in the hand, which showed no statistically significant difference. For the hand, there was a shift from peak frequency to lower frequencies, while for the forearm, the response to stimulation was the opposite. This corroborates the idea that mechanical stimulation can have different effects on different limbs when they are simultaneously stimulated by the same stimulation pattern.

This result suggests that the inter-segment peripheral mechanical stimulus response may be influenced by the frequency pattern or even by the stimulus generation region. For example, in this study, there were more vibrotactile stimulation pathways for the hand than for the forearm (Figure 3), as all fingers and the palm received mechanical stimulation. On the other hand, the forearm received only one stimulation pathway located at approximately one third of the wrist joint.

In addition, the entrainment of the peak frequency may be indicative that the 250 Hz stimulation influenced the central oscillations of the tremor. This would support the hypothesis proposed in this study that the introduction of vibration can induce oscillation capable of redefining the central oscillations associated with tremor. However, this difference in tremor frequency does not appear to be sufficient to fully explain the hypothesis of this study.

Vibrotactile stimulation altered not only the main component of the tremor (the one with the highest power) but also the adjacent components with lower power, as illustrated in Figure 18, Figure 19 and Figure 20. Not only the 250 Hz stimulus but also the other stimuli modified these components, to a greater or lesser degree, in different ways. For instance, certain stimulation patterns led to a reduction in the low-frequency components, even though they did not decrease or may have even increased the main component of the tremor. In this regard, additional research on the impact of vibrotactile stimulation on particular tremor components is required, because this may contribute to differential diagnosis or perhaps the development of alternative treatments.

The changes in the amplitude, peak power, peak frequency, and regularity of the tremor may indicate a positive effect of the 250 Hz vibratory stimulation on tremor suppression. However, this effect was limited for a short period of time, for instance, approximately one minute after stimulation. This limited effect may be due to the employed experimental protocol, which did not consider an increased period of analysis after stimulation, leading to pertinent questions. How long does the positive effect of vibrotactile stimulation on tremor persist? Is there an optimal duration or time interval during which stimulation leads to tremor stabilisation, with regard to amplitude and regularity? Can extended periods of stimulation induce motor fluctuations?

### 4.4. On Signal Processing Stages

The signal processing stages proposed (Figure 5) in this research can be applied to the analysis of any type of tremor, and, more importantly, they can deal with the noise produced by the stimulus by attenuating it from the recorded signal (Figure 9). The analysis was designed to provide an objective evaluation of the tremor from distinct but complementary perspectives, i.e., the understanding of the effect of the stimulus on the tremor amplitude, which is more clinically intuitive, as well as the analysis of the tremor characteristics in the frequency domain and the way the stimulation may affect tremor regularity.

This type of analysis is important because tremor amplitude determines the symptomaticity of the condition. In clinical practice, its analysis is more common. This is because tremor amplitude is easier to assess through observation alone, as it is included in several clinical scales such as the FTM [39] and TETRAS [40]. However, these scales have limitations when it comes to assessing variations in tremor amplitude over time and the fact that they assign different scores to different amplitude ranges.

Moreover, the other tremor characteristics (frequency, power, and entropy) carry relevant information that cannot be perceived and discriminated with the naked eye [45]. In this sense, the understanding of the behaviour of these inherent tremor characteristics may contribute to an improved characterisation of tremor, the effect of external inputs on its generating mechanisms, and even its discrimination according to the disorder it is associated with.

### 4.5. Limitations of the Experimental Protocol and Future Perspectives

Figure 4 depicts the experimental protocol adopted in this research. This protocol was designed with the intention of verifying possible differences in the tremor pattern before, during, and after stimulation, considering different stimulation patterns [13]. Despite several interesting aspects of this protocol, a limiting aspect is that it restricts the period of stimulation and pre- and post-stimulus evaluation to fixed values without accounting for possible variation among the participants, especially after stimulation.

For some participants, the effect of the stimulus may be more acute, meaning that it can be seen just after stimulation, whereas for others, this effect might appear over a longer period, meaning that based on a fixed time protocol, one could not detect it (Figure 10). The visual analysis of the signals suggested that for some individuals, there is a latency in the effect of the stimulation, further suggesting the need for customised experimental protocols that can take into account the time response of the individual.

Considering the limitations of the experimental protocol, some modifications could be implemented in future research. First, it would be advisable to incorporate a calibration stage for the evaluation of the individual response time to the vibrotactile stimulus. This calibration stage would adopt the use of relative time periods ranging from 0 to 100%, facilitating comparisons of individuals’ responses independently of fixed time intervals.

Second, introducing a random presentation of the stimulating patterns is pertinent because one pattern may influence the response to another. Third, it is important to consider the effect of rest time on the stimulation period. Is 10 min enough to suppress the effect of a previous stimulus? Fourth, given the distinct effects of stimulating the hand and the forearm, it may be necessary to include the possibility of simultaneously stimulating distinct body parts with different stimulating patterns.

Lastly, a relevant aspect to observe is with regard to the tolerance of the subject to the vibrotactile stimulus, as one can have tremor dampened by a specific stimulating pattern; however, the stimulus could cause pain or any other discomfort to the person. Additionally, individual responses to vibrotactile stimulation can vary, as they do for other modalities (e.g., vision, hearing, and touch) [37,38]. Therefore, the same intensity of stimulation may not be equally effective for everyone.

The findings of this study open up several interesting issues for future research. One potential direction is to deepen our understanding of the effect of vibrotactile stimulation on the underlying components of tremor in ET. In this context, studies combining analysis with inertial sensors and electromyography could provide additional insights into the effect of vibrations on the components of tremor.

Furthermore, increasing the sample size by forming homogeneous groups based on the level of severity of the disorder could provide valuable information on the clinical applicability of the findings. Additionally, efforts should be directed towards enhancing the developed regression model with this balanced dataset and integrating the model into a real-time system for controlling the frequency of vibrotactile stimulation. This could lead to new technologies for tremor damping.

## 5. Conclusions

The study verified that peripheral vibrotactile stimulation is a potential way to noninvasively change the dynamics of postural tremor in individuals with ET. Among the various stimuli investigated, the 250 Hz vibratory stimulus had the most pronounced positive effect on the motor response of individuals with ET during posture maintenance. Therefore, mechanical vibrations have the potential to be clinically used in the field of tremor management while also offering valuable insights into the dynamics of the tremor in response to external inputs. However, the clinical application of vibratory stimuli requires further studies on the duration of the motor response to vibration in individuals with ET. The results are limited because of the reduced number of participants in the research; thus, future studies should increase the sample size to account for inherent variability among severity levels of tremor in individuals with ET.

## Figures and Tables

**Figure 1 healthcare-12-00448-f001:**
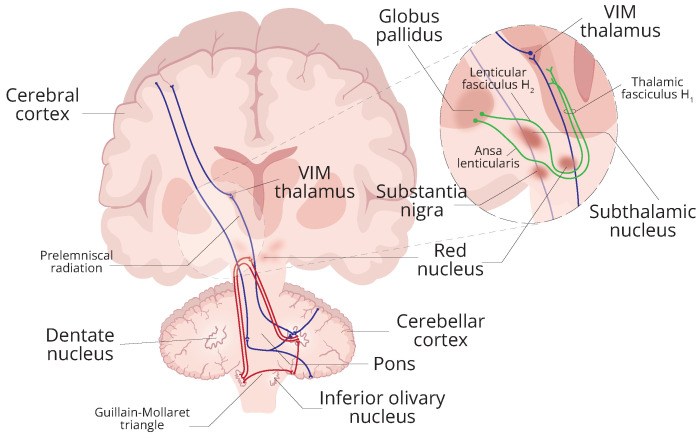
Simplified representation of pathways involved in the pathophysiology of ET. In red, the Guillain–Mollaret triangle loop is shown. In this pathway, the dentate nucleus connects to the contralateral red nucleus through the superior cerebellar peduncle, the red nucleus to the inferior olivary nucleus (ION) through the central tegmental tract, and the ION to the contralateral dentate nucleus through the inferior cerebellar peduncle. Some dentate fibres form synapses in the red nucleus (in red), while others circumnavigate it (in green), forming important projections between the thalamus and globus pallidus. It is an important feedback circuit of the brainstem and deep cerebellar nuclei that regulates spinal cord motor activity. In dark blue, the cortico–ponto–cerebello–thalamo–cortical loop is shown. The cortico–ponto–cerebellar pathway connects the cerebrum to the cerebellum through the pons and contralateral middle cerebellar peduncle. The cerebello–thalamo–cortical pathway connects the cerebellum to the cerebrum through the superior cerebellar peduncle and the contralateral thalamus. The cortico–ponto–cerebello–thalamo–cortical loop plays a role in motor control.

**Figure 2 healthcare-12-00448-f002:**
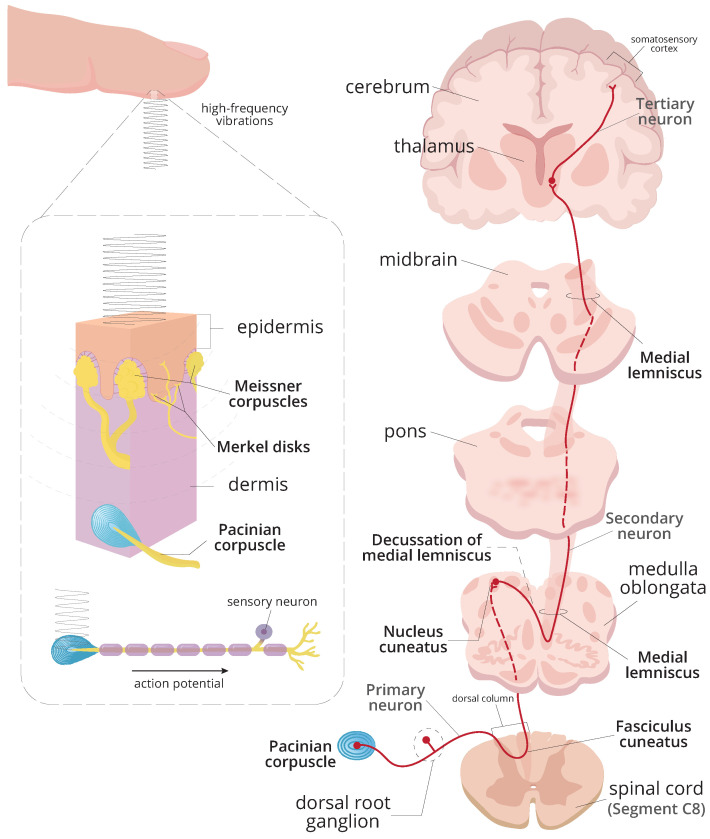
Simplified representation of the vibratory pathway. When a high-frequency vibrotactile stimulus is applied to the skin surface of the fingertip, the mechanical vibrations penetrate the layers of the skin and are sensed by peripheral receptors known as Pacinian corpuscles. If the intensity of the vibratory stimulus is sufficient to reach or exceed the threshold, an action potential is produced and propagated towards the central nervous system. When the brain interprets the vibratory sensation that reaches the cortex, the perception of the stimulus occurs. Thus, appropriate motor responses are generated based on the stimulus received.

**Figure 3 healthcare-12-00448-f003:**
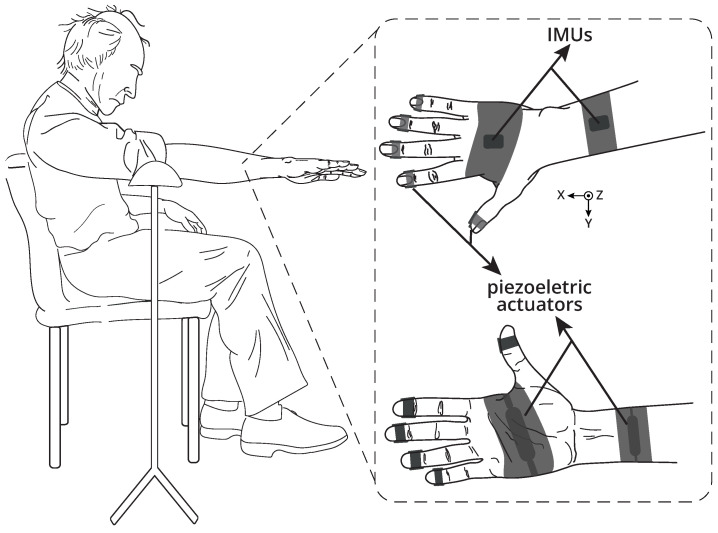
Illustration of the experimental scenario. The participants sat in a chair with their feet flat on the floor and their backs against the backrest. The arms were maintained parallel to the transverse plane and 90° from the coronal plane. Inertial sensors (IMUs) were positioned at the back of the hand and forearm. Piezoelectric actuators were placed on the fingertips, hand, and forearm.

**Figure 4 healthcare-12-00448-f004:**
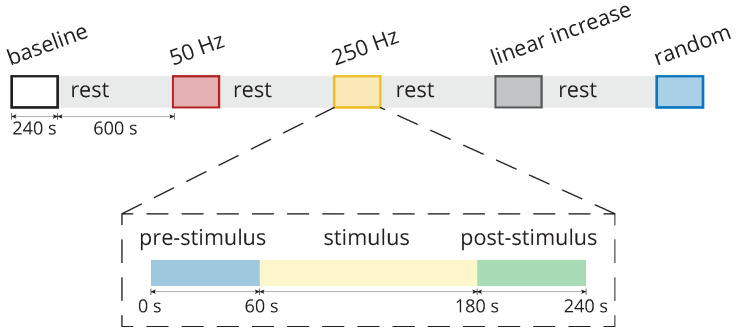
Each individual was subjected to one of the five stimulation paradigms during posture maintenance: baseline (without vibrotactile stimulation), stimulation at 50 Hz, stimulation at 250 Hz, stimulation with increasing linear frequency, and random frequency stimulation. All stimulation paradigms lasted 4 min, and a 10 min rest period followed each one. Except for the baseline, the stimulation paradigms contained a pre-stimulus region lasting 60 s, a stimulus region lasting 120 s, and a post-stimulus region lasting 60 s.

**Figure 5 healthcare-12-00448-f005:**
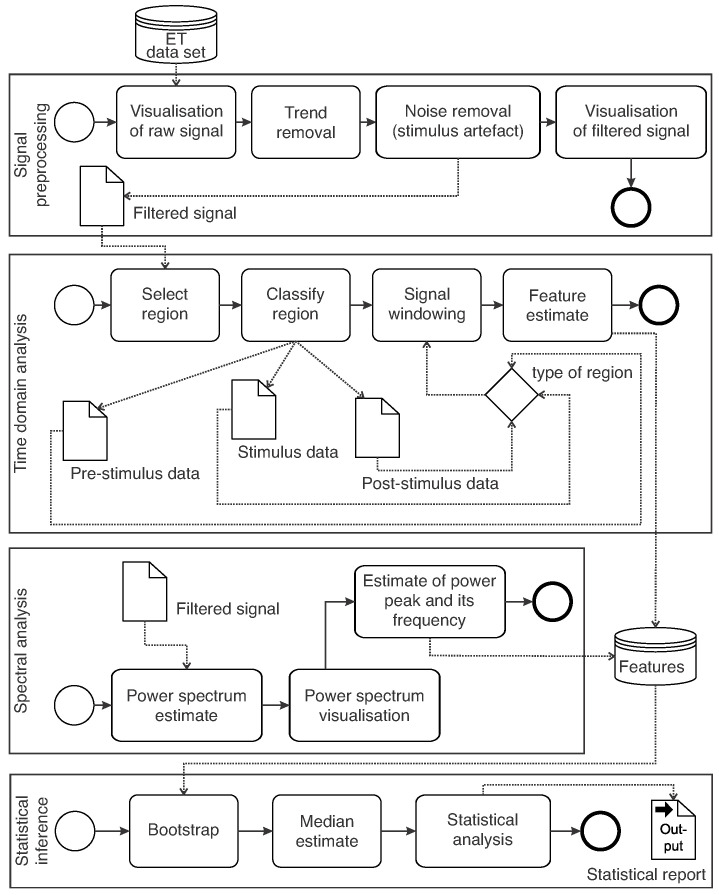
Signal processing steps for feature extraction and statistical analysis.

**Figure 6 healthcare-12-00448-f006:**
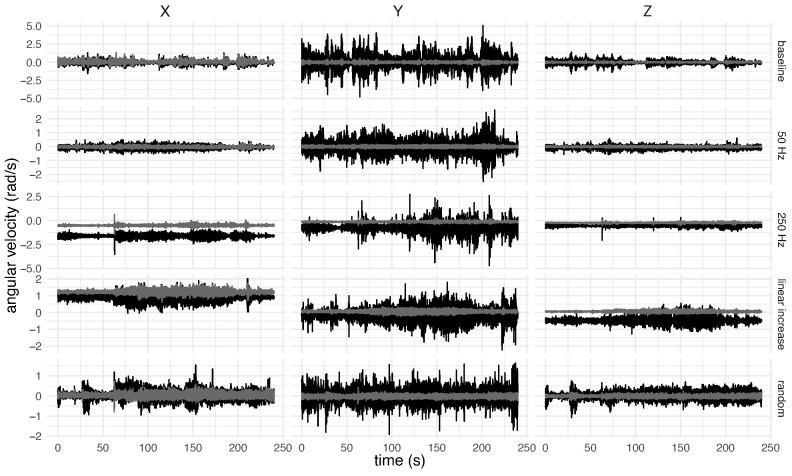
Typical signals were collected for each axis and stimulation paradigm. The signals in black are from the sensor positioned at the hand, whereas the signals in grey are from the sensor positioned at the forearm.

**Figure 7 healthcare-12-00448-f007:**
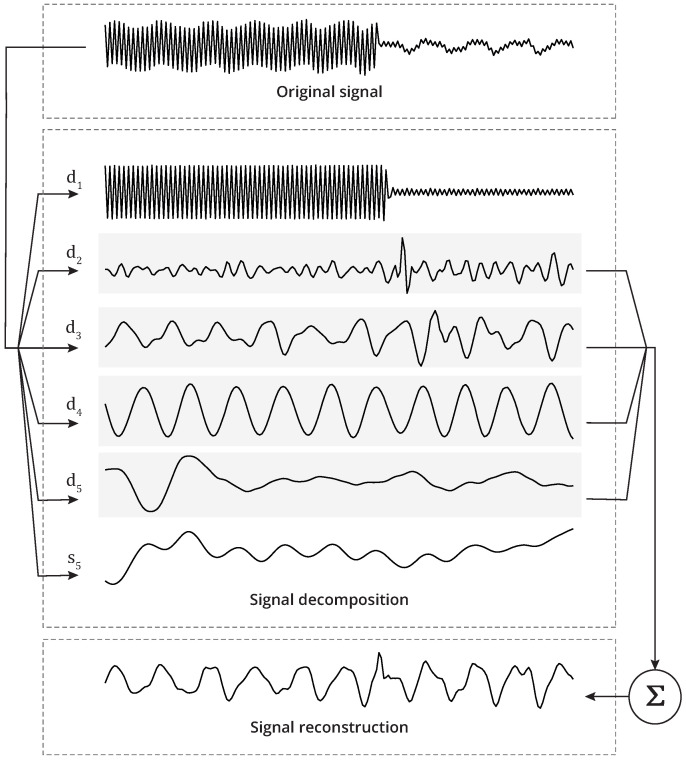
The strategy for signal filtering using wavelets.

**Figure 8 healthcare-12-00448-f008:**
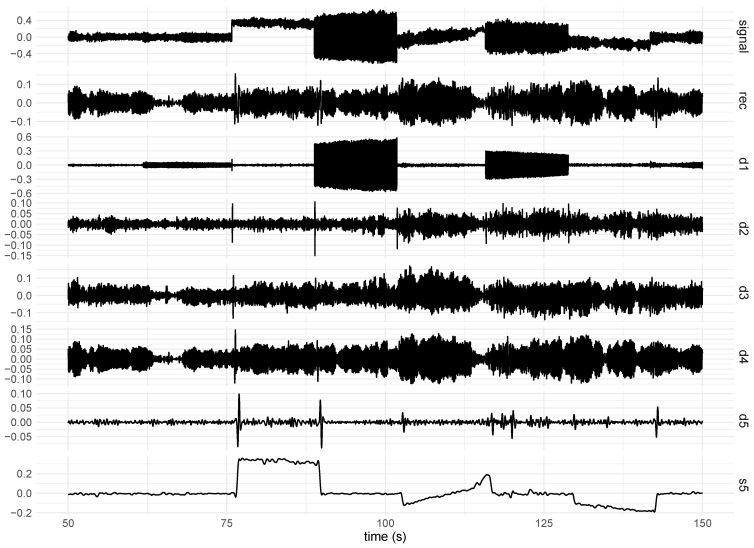
An example of a signal acquired using a gyroscope positioned on the subject’s hand. The signal was collected while the subject received a linearly increasing frequency stimulus. The detail components from d1 to d5 are shown along with the approximation component (s5). The filtered signal (rec) resulting from the summation of the components from d1 to d5 is shown.

**Figure 9 healthcare-12-00448-f009:**
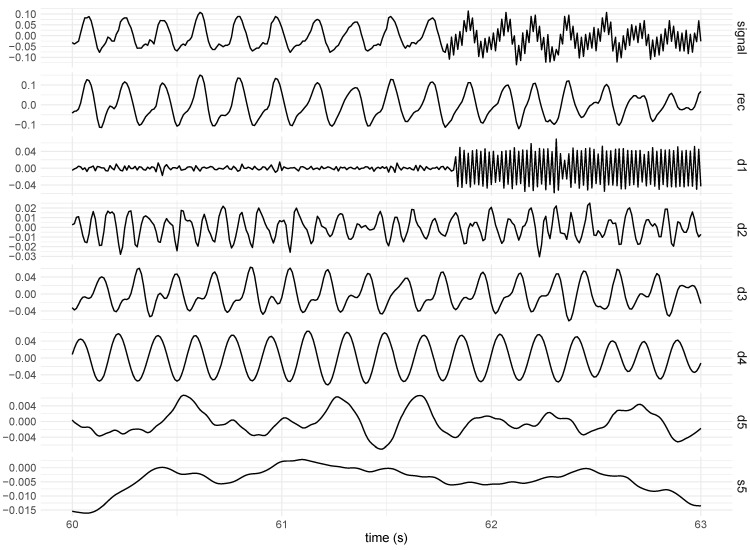
An example of a signal acquired using a gyroscope positioned on the subject’s hand. The signal was collected while the subject received a linearly increasing frequency stimulus. The influence of the stimulus on the signals can be seen from 61.78 to 63.00 s. Detail component d1 is clearly related to the noise originating from the stimulus. Thus, the reconstructed signal (rec) was obtained by removing d1 along with the approximation component (s5), which was associated with the signal trends.

**Figure 10 healthcare-12-00448-f010:**
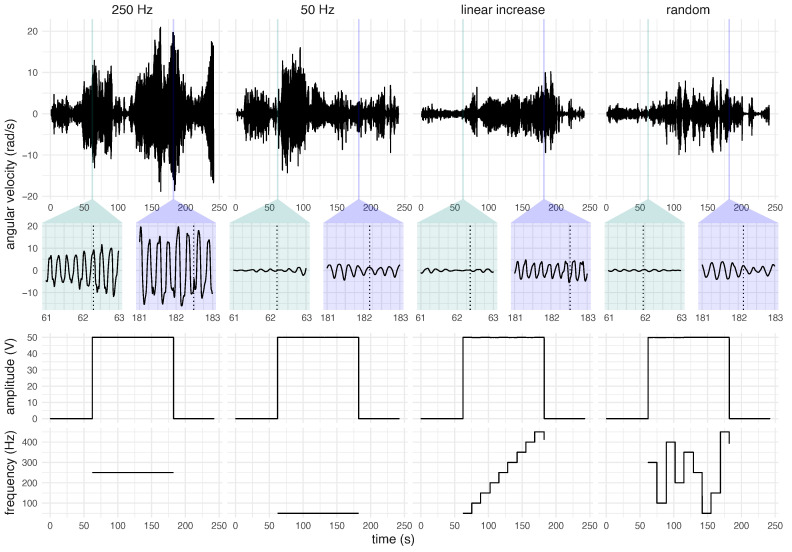
Typical angular velocity (rad/s) in response to various stimulation patterns (50 Hz, 250 Hz, linear increase, and random). In each signal, two regions were highlighted: the first is the transition between periods without and with stimulation (green area), and the second is the transition between periods with and without stimulus (blue area). Pairs of vertical dashed lines indicate the beginning (green area) and end (blue area) of the period of stimulation. The controlling signals shown in the third row of the figure determine the stimulus region. At the bottom, there are distinct stimulus patterns.

**Figure 11 healthcare-12-00448-f011:**
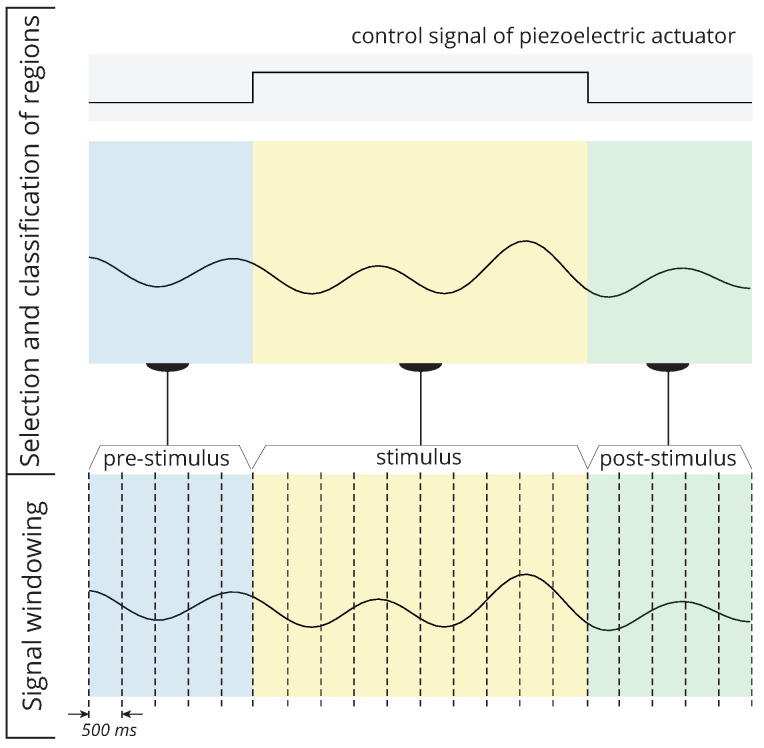
The processes required to window the raw signals are depicted. First, the signals are classified into regions based on the presence or absence of stimulus. Second, for feature extraction, the signal in each region is segmented into 500 ms windows.

**Figure 12 healthcare-12-00448-f012:**
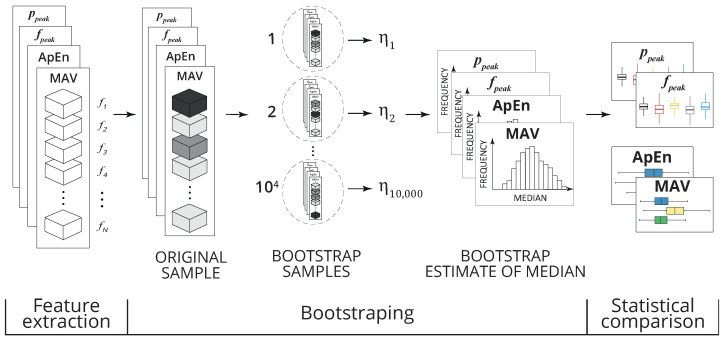
An overview of statistical analysis. From the original samples, 10,000 bootstrap samples were estimated in total. The median of each bootstrap sample was calculated, and the distribution of the median was visualised. The boxplot shows the median of each feature.

**Figure 13 healthcare-12-00448-f013:**
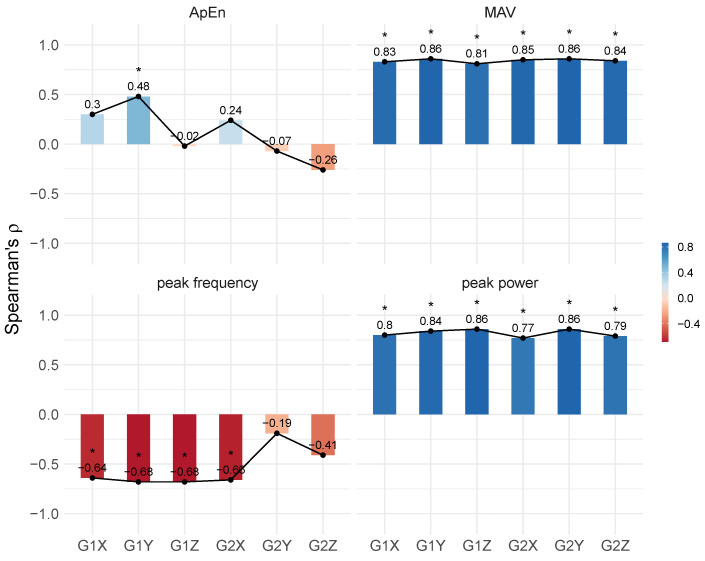
Relationship between clinical and instrumental assessment. Significant correlations are indicated by asterisks (*p*-value <0.05).

**Figure 14 healthcare-12-00448-f014:**
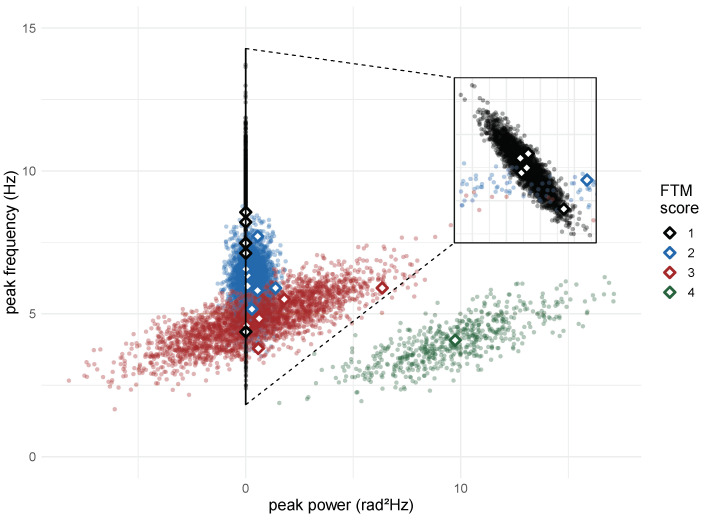
Relationship between peak frequency and peak power for each clinical score. Each level of the clinical scale is represented by a different colour. The cloud of points, represented by circles, corresponds to the augmented data, whereas the diamonds correspond to the original data. In addition, the region comprising the distribution of data for score 1 (in black) is shown in the inset.

**Figure 15 healthcare-12-00448-f015:**
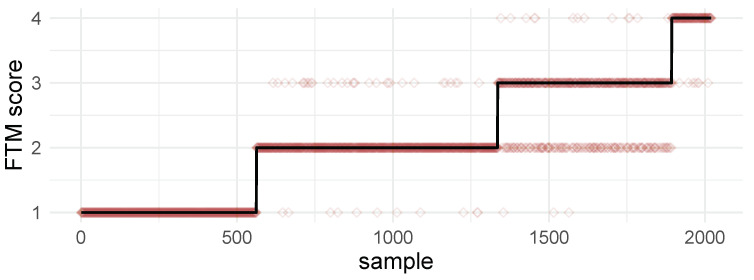
Predicted and original disease severity of the testing set. The black curve corresponds to the expected severity scores, whereas the points (in red) correspond to the scores predicted by the proposed model. When the predicted scores follow the expected output, there is an overlap between the black line and the estimated data points in red.

**Figure 16 healthcare-12-00448-f016:**
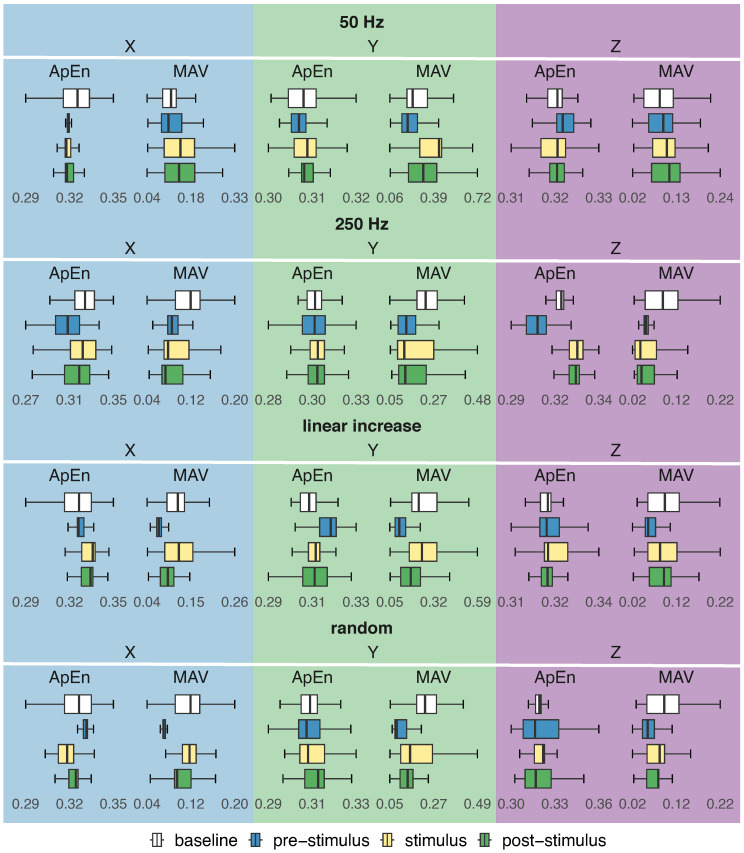
Boxplots comparing the amplitude and regularity of the involuntary activity in the absence (baseline) and presence of the vibrotactile stimulation in the pre-stimulus, stimulus and post-stimulus regions for all stimulation patterns (the frequency of 50 Hz, 250 Hz, increasing linear and random) and each axis (X, Y and Z) from the sensor positioned at the hand. In the boxplots, the central horizontal line indicates the median, box boundaries indicate the first and third quartiles, and the whiskers indicate the minimum and maximum values. Extreme values were removed from the analysis.

**Figure 17 healthcare-12-00448-f017:**
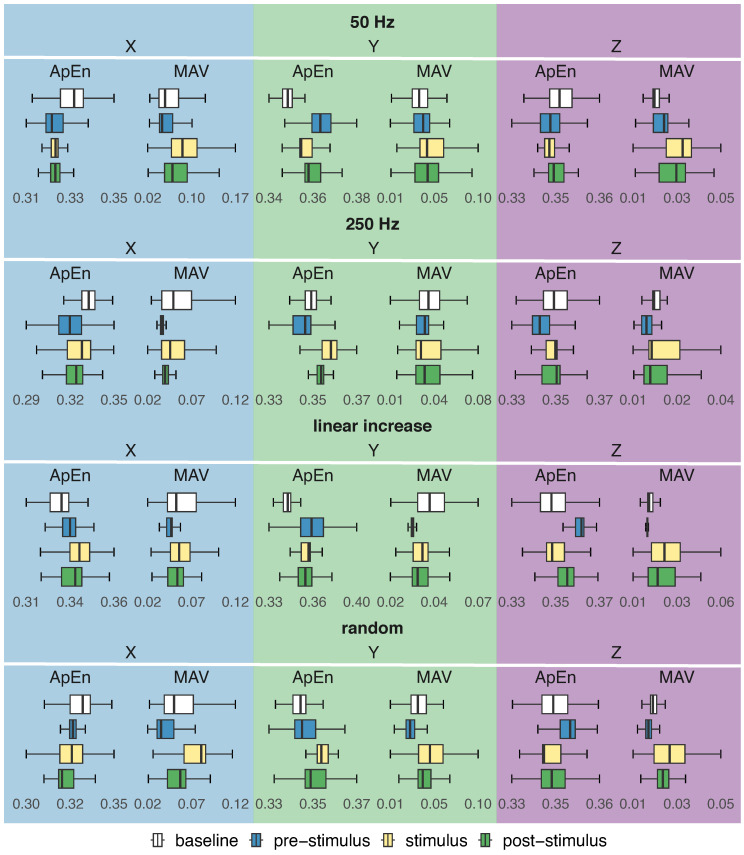
Boxplots comparing the amplitude and regularity of the involuntary activity in the absence (baseline) and presence of the vibrotactile stimulation in the pre-stimulus, stimulus and post-stimulus regions for all stimulation patterns (the frequency of 50 Hz, 250 Hz, increasing linear and random) and each axis (X, Y and Z) from the sensor positioned at the forearm. In the boxplots, the central horizontal line indicates the median, box boundaries indicate the first and third quartiles, and the whiskers indicate the minimum and maximum values. Extreme values were removed from the analysis.

**Figure 18 healthcare-12-00448-f018:**
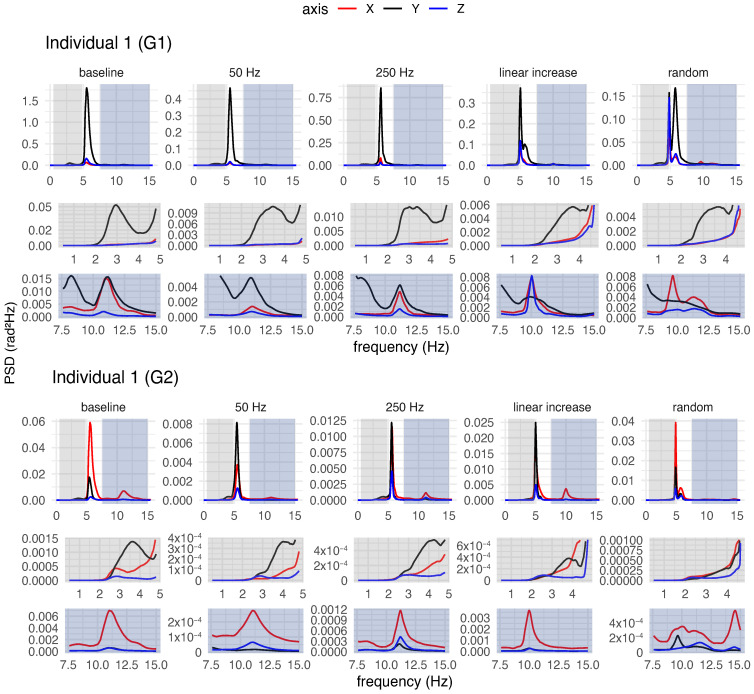
Power spectral densities of the angular velocity of the hand (G1) and forearm (G2) of Individual 1 for each axis (X, Y and Z) during posture maintenance, without (baseline) and with (50 Hz, 250 Hz, linear and random increase) vibrotactile stimulation. The highlighted areas of the spectra (the grey and blue windows) depict the spectral behaviour for lower and higher frequency bands in relation to the peak frequency.

**Figure 19 healthcare-12-00448-f019:**
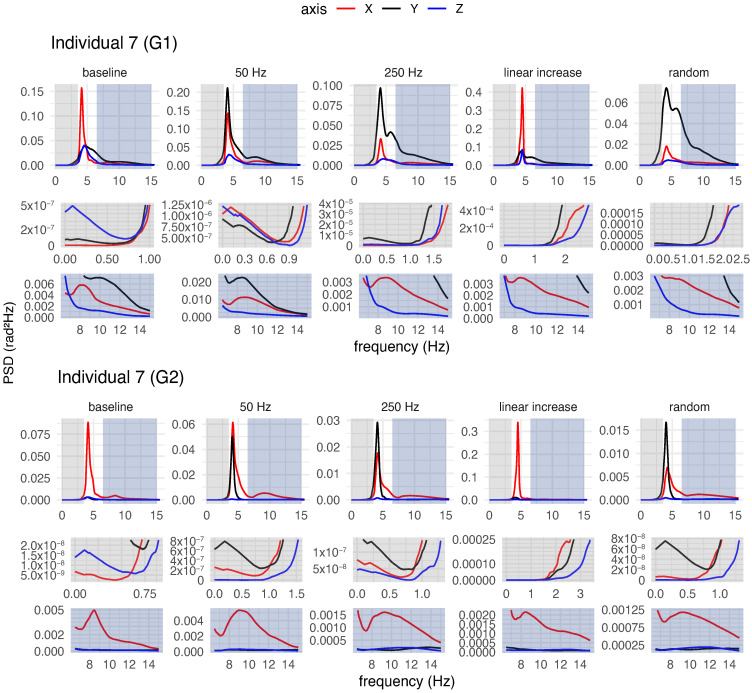
Power spectral densities of the angular velocity of the hand (G1) and forearm (G2) of Individual 7 for each axis (X, Y and Z) during posture maintenance, without (baseline) and with (50 Hz, 250 Hz, linear and random increase) vibrotactile stimulation. The highlighted areas of the spectra (the grey and blue windows) depict the spectral behaviour for lower and higher frequency bands in relation to the peak frequency.

**Figure 20 healthcare-12-00448-f020:**
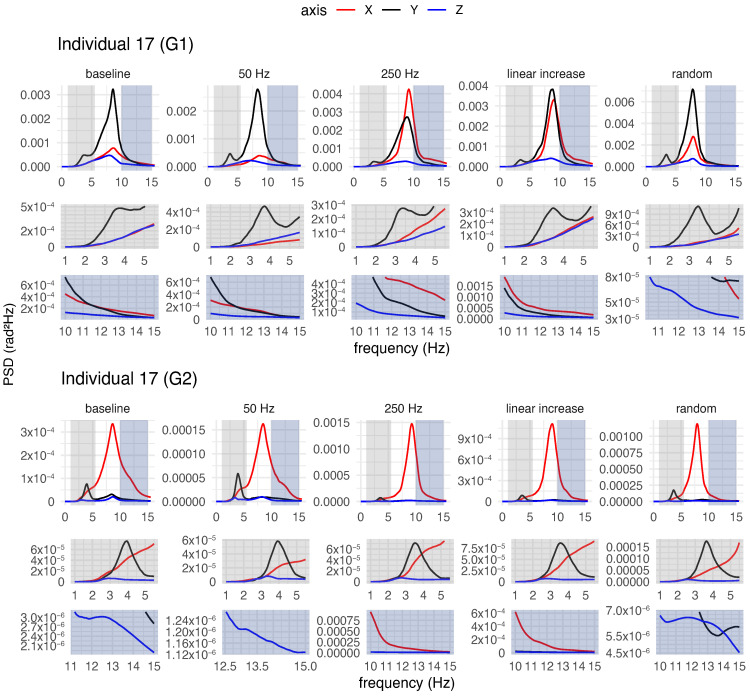
Power spectral densities of the angular velocity of the hand (G1) and forearm (G2) of Individual 17 for each axis (X, Y and Z) during posture maintenance, without (baseline) and with (50 Hz, 250 Hz, linear and random increase) vibrotactile stimulation. The highlighted areas of the spectra (the grey and blue windows) depict the spectral behaviour for lower and higher frequency bands in relation to the peak frequency.

**Figure 21 healthcare-12-00448-f021:**
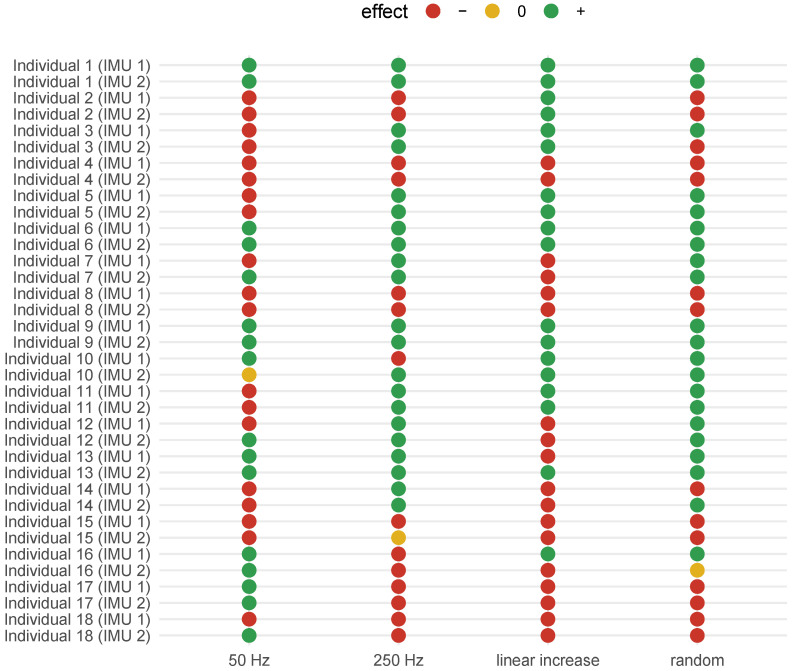
Effect of the different stimulation patterns on the involuntary activity of individuals with ET. In this representation, each row corresponds to an individual (for both placements of the IMUs) and each column corresponds one type of the vibrotactile stimulus. The colours represent changes in the spectral magnitude of the tremor during each upper limb stimulation. Red means the stimulation pattern caused an increase in maximum power (denoted by −, i.e., a negative effect), yellow indicates that the stimulation pattern had no noticeable effect (denoted by 0, i.e., a neutral effect), and green means that the stimulation pattern caused a reduction in maximum power (denoted by +, i.e., a positive effect).

**Figure 22 healthcare-12-00448-f022:**
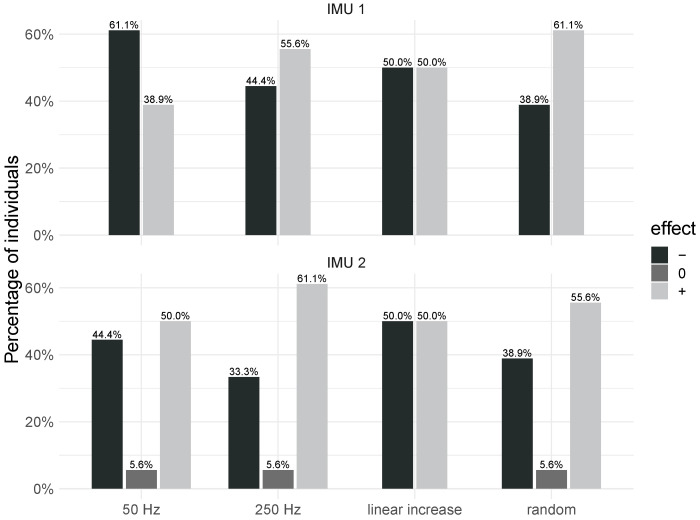
Grouped barplots comparing the percentage of individuals whose the involuntary hand (IMU 1) and forearm (IMU 2) activity presented or did not present a response to different stimulation patterns during posture maintenance. In the barplots, each rectangular bar represents the proportion of individuals who showed an increase in the maximum power (denoted by −, i.e., a negative effect), a decrease in power (denoted by +, i.e., a positive effect), or no noticeable effect (denoted by 0, i.e., a neutral effect) when the upper limb was stimulated for each stimulation pattern (stimulation at 50 Hz, 250 Hz, increasing linear frequency, and random frequency).

**Figure 23 healthcare-12-00448-f023:**
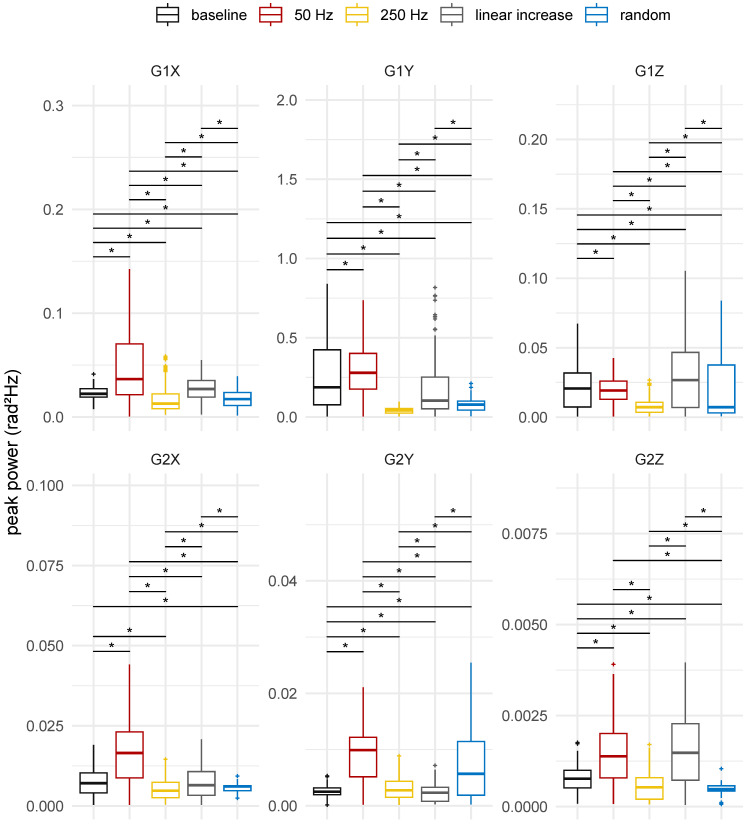
Boxplots comparing the peak power of involuntary activity in the presence (the frequency of 50 Hz, 250 Hz, increasing linear and random) and absence (baseline) of the vibrotactile stimulation for each axis (X, Y and Z) and sensor positioning (1 corresponding to the sensor positioned on the hand and 2 corresponding to the sensor positioned on the forearm). In the boxplots, the central horizontal line indicates the median, box boundaries indicate the first and third quartiles, the whiskers indicate the minimum and maximum values, and the plus signs indicate the outliers. Extreme values were removed from the analysis. For each axis and sensor positioning, significant pairwise comparisons between the trials with and without stimulation are denoted by an asterisk, considering the pairwise Wilcoxon rank-sum test using the Bonferroni correction to identify which groups were different.

**Figure 24 healthcare-12-00448-f024:**
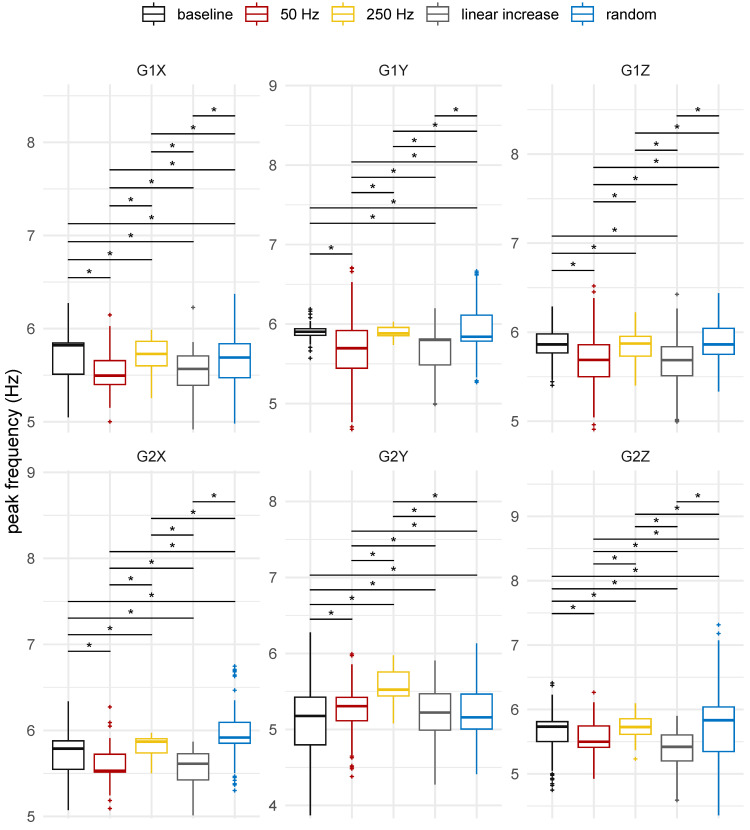
Boxplots comparing the peak frequency of involuntary activity in the presence (the frequency of 50 Hz, 250 Hz, increasing linear and random) and absence (baseline) of the vibrotactile stimulation for each axis (X, Y and Z) and sensor positioning (1 corresponding to the sensor positioned on the hand and 2 corresponding to the sensor positioned on the forearm). In the boxplots, the central horizontal line indicates the median, box boundaries indicate the first and third quartiles, the whiskers indicate the minimum and maximum values, and the plus signs indicate the outliers. Extreme values were removed from the analysis. For each axis and sensor positioning, significant pairwise comparisons between the trials with and without stimulation are denoted by an asterisk, considering the pairwise Wilcoxon rank-sum test using the Bonferroni correction to identify which groups were different.

## Data Availability

Data are contained within the article and Appendix A.

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
