# Peer review of "On the Effect of Vibrotactile Stimulation in Essential Tremor"

_healthcare, 2024, doi:10.3390/healthcare12040448_

Round 1

Reviewer 1 Report

Comments and Suggestions for Authors

The authors studied the effect of vibrotactile stimulation on essential tremor, using a paradigm that evaluated short-term efect on the tremor. The authors noted that 250 Hz stimulation reduced the amplitude of tremor. While the protocol precluded the assessment of long-term benefits, the study can inspire further research and has the potential to lead to novel treatment of essential tremor.

The authors provide only a limited introduction to another form of treatment:  neuromodulation therapy by electric stimulation (there is already an FDA-cleared device). Reference to the study by Isaacson et al. Prospective Home-use study on Non-invasive Neuromodulation Therapy for Essential Tremor. Tremor Other Hyperkinet Mov (NY). 2020;10:29 may be appropriate.

Author Response

Please, see the attached document. 

Reviewer 2 Report

Comments and Suggestions for Authors

Following the hypothesis that peripheral vibratory stimulation may relieve upper limb postural tremor by inhibiting the cuneate nucleus, the authors here assessed the effects of specific patterns of vibrotactile stimulation on tremor amplitude, regularity and energy, in 18 patients with essential tremor (ET). More in detail, five vibrotactile stimulation paradigms were administered using specific piezoelectric actuators: sham, constant frequencies at 50 and 250 Hz, and lastly, alternating patterns modulated by increasing frequency between 50 and 450 Hz with resolution of 50 Hz, randomly elicited. Baseline tremor activities as well as vibrotactile-induced modified patterns of tremor were recorded through inertial measurement units (IMUs). Data used for the following statistic analysis mainly based on spectral analysis was extracted from the Lora-Millán dataset. The authors found that at baseline, the upper limb tremor in ET patients is neurophysiologically characterized by several peaks of oscillatory activity ranging from 4 to 12 Hz and that each of these peaks can be explained by the synchronous activity of specific networks, from those of physiologic orthostatic tremor to pathological networks. Then, the authors identified specific effects in terms of amplitude, frequency and entropy modulation by applying four patterns of vibrotactile stimulation: briefly, they reported the most relevant and beneficial effect on tremor when using the 250 Hz frequency. 

This study may be, in my opinion, potentially relevant since the results outlined here would imply innovative therapeutic strategies in ET. Also, the spectral analysis may help to unravel the pathophysiological underpinnings of tremorous activities in ET. However, I have several concerns about the study, mainly regarding clinical issues. 

The sample size is rather small and it seems a prior analysis for identifying the optimal sample size has not been implemented by the authors. As reported, ET is one of the most frequent movement disorders, I believe it would not be so though to increase the sample accordingly. Related to the issue, the cohort of participants is rather heterogeneous and the heterogeneity in baseline peaks of frequency among ET patients may be related to that. A possible hypothesis for refining the results would imply a cluster analysis of ET patients manifesting similar patterns of clinical tremor.

Correlations between clinical and instrumental measures are missing.

No insight into the pathophysiology of tremors has been provided. For instance, the role of the cuneate nucleus that has been suggested in the introduction to be involved in the pathogenesis of vibratory-dependent tremor relief has not been further expanded in the discussion.

Comments on the Quality of English Language

There are several typos all over the manuscript that should be carefully addressed and revised by the authors. I would suggest extensive English editing. 

Author Response

Please, see the attached document. 

Reviewer 3 Report

Comments and Suggestions for Authors

The article titled "Effect of Peripheral Vibrotactile Stimulation on Postural Tremor in Individuals with Essential Tremor" investigates how different patterns of vibrotactile stimulation affect postural tremor in individuals with essential tremor (ET). The study was conducted with 18 participants using inertial sensors placed on the hand and forearm. Four patterns of stimulation were evaluated: 50 Hz, 250 Hz, linear frequency increase, and random frequency.

The methodology appears adequate, with a clear and detailed focus on participant selection, stimulation protocol, and data analysis. However, there are aspects that could be improved:

Methodology and Presentation of Results:

The sample of only 18 participants might limit the generalization of the results. A larger sample size could increase the robustness of the findings.

The authors might consider including a control group to strengthen the validity of the results.

Conclusions:

The conclusions are well aligned with the obtained results. However, the authors should be cautious not to overestimate the clinical applicability of their findings given the limited sample and the absence of a control group.

Limitations:

The study acknowledges its limitations, such as the short post-stimulation follow-up period and individual variability in response to stimulation. These limitations are relevant and are adequately addressed, suggesting areas for future research.

Although these limitations and issues are present, the article mentions and carefully considers them in the discussion section. It has a very good initial approach to establishing a protocol for treating essential tremor using a wearable system.

I consider it acceptable for publication.

Author Response

Please, see the attached document. 

Reviewer 4 Report

Comments and Suggestions for Authors

The paper outlines a study on the effects of vibrotactile stimulation on tremor dynamics in individuals with Essential Tremor (ET). In their work, the authors see the potential for electrical stimulation in the upper limb to induce intense normal physiological stimulation (inhibitory), which can reduce pathological oscillations and may therefore reduce the tremor observed in ET patients. The work is interesting and everything related to is a progressive and chronic neurological disorder. In addition, there is a kind of revival of electrical stimulation in the treatment of neurological diseases. However, I have some comments and questions for this work:

Point 1. Since the Abstract is the presentation part of the work, everything must be presented there clearly and not interfere with the extended information presented later in the article. If the database used in this research originated from a previous study, this must be reflected in the abstract. Methods and conclusions do not reflect the manuscript context.

Point 2. Since this is an ongoing study, it is necessary to define very clearly what has been done with these data before what results, and what new questions and scientific challenges are solved in this manuscript.

Point 3. If the database used in this research originated from a previous study, then I think there is no need to describe and repeat everything (in another form) in the methodology, but it is enough to present only the main aspects most important to the new study and connect them with new scientific insights and possibly already performed, obtained results. I think the authors should consider reorganizing the methodological part. It is enough to cite methodological aspects from the previous published work.

Point 4. The results section is very difficult to follow because there is still a lot of methodological information, not only the results are presented, but also the data processing is described. In the Results section, there are further discussion considerations. In addition, the figures e.g. 11 and 12 boxplots are presented in a different scale, away from the text describing them. Authors should avoid redundant information in the results section and present only the results in a form that is as simple and understandable as possible for the reader. Now there is a lot of information, it becomes impossible to follow or compare it.

Point 5. I think that the authors do not need to show the whole "kitchen" of the study. Because it is difficult to choose which information is related to the research goals and which is just a step toward the goal. It complicates the work, it is easy to get lost in it.

Point 6. Limitations are relevant, but they are repeated from the previous study from which the data are taken. The authors could indicate more precisely, and discuss why they want to present the data of only three individuals, and why they do not try to find homogeneous groups, average them, etc.

Point 7. Compared to the presented obvious conclusions, the work seems somewhat "inflated".

Point 8. When it is mentioned everywhere that the work will be continued, then maybe answers could be given:

·         What are the potential next steps or future research directions based on the findings of this study?

·         Are there specific challenges or unanswered questions that emerged during the study that could guide future investigations?

Author Response

Please, see the attached document. 

Round 2

Reviewer 2 Report

Comments and Suggestions for Authors

No further comments toward the study.

Comments on the Quality of English Language

Minor english spelling flaws. Please revise altogheter. 

Author Response

Please, see the attached document.

Reviewer 4 Report

Comments and Suggestions for Authors

The publication has been significantly improved and supplemented. All comments have been answered. Thank you to the authors for their hard work and goodwill. Now the material is much clearer, but maybe those pictures that take up more than half a page can be added to the appendices or some can even be removed.

Overall, I think it will be a great publication for an interested audience.

Author Response

Please, see the attached document.
